

# Emulating the future distribution of perennial firn aquifers in Antarctica

Sanne B. M. Veldhuijsen[1], Willem Jan van de Berg[1], Peter Kuipers Munneke[1], Nicolaj Hansen[2], Fredrik Boberg[2], Christoph Kittel[3,4], Charles Amory[3], and Michiel R. van den Broeke[1]

[1]Institute for Marine and Atmospheric Research Utrecht, Utrecht University, Utrecht, The Netherlands
[2]Department of National Centre for Climate Research (NCKF), Danish Meteorological Institute, Copenhagen, Denmark
[3]Institut des Géosciences de l'Environnement (IGE), Univ. Grenoble Alpes/CNRS/IRD/G-INP, Grenoble, France
[4]Département de Géographie, SPHERES, ULiège, Liège, Belgique

**Correspondence:** Sanne B. M. Veldhuijsen (s.b.m.veldhuijsen@uu.nl)

**Abstract.** Perennial firn aquifers (PFAs) are year-round bodies of liquid water within firn, which modulate meltwater runoff to crevasses, potentially impacting ice-shelf and ice-sheet stability. Recently identified in the Antarctic Peninsula (AP), PFAs form in regions with both high surface melt and snow accumulation rates, and are expected to expand due to the anticipated increase in melt and snowfall. Using a firn model to predict future Antarctic PFAs for multiple climatic forcings is computationally

expensive. To overcome this, we developed an XGBoost emulator, a fast machine learning model, to approximate a firn model. The PFA emulator was trained with simulations from the firn densification model IMAU-FDM, forced by three emission scenarios (SSP1-2.6, SSP2-4.5 and SSP5-8.5) of the combined regional climate model (RCM) RACMO2.3p2 and general circulation model (GCM) CESM2. Using a scenario and spatial blocking evaluation approach, we found that the emulator successfully explains at least 89 % of PFA presence and meltwater storage variance. Using the PFA emulator, we predict

future PFAs (2015-2100) for nine additional forcings from the RCMs MAR and HIRHAM in combination with five GCMs. Under SSP1-2.6 and SSP2-4.5, PFAs remain mostly restricted to the AP. For SSP5-8.5, PFAs expand to Ellsworth Land in West Antarctica, and Enderby Land in East Antarctica. For climatic forcings from RACMO and MAR, we find that liquid water input (melt and rain) and snow accumulation are good predictors for PFA occurrence. However, HIRHAM predicts considerably less surface melt and accumulation for a given temperature than MAR and RACMO do, resulting in less realistic PFA predictions.

Overall, our findings show that PFAs will likely expand in a warmer Antarctica, irrespective of the emission scenario.

## 1 Introduction

The Antarctic Ice Sheet (AIS) has been losing mass since at least 2002 (Shepherd et al., 2019), contributing $\sim$ 10 % to global average sea level rise since 1993 (Oppenheimer et al., 2019). The primary factor currently contributing to AIS mass loss is enhanced basal melt beneath ice shelves (Smith et al., 2020), leading to their thinning, enhanced iceberg calving or collapse.

This, in turn, reduces their buttressing effect, allowing inland ice to flow faster into the ocean. Another process to reduce the buttressing effect of ice shelves is melt pond-driven hydrofracturing, which is expected to increase under future warming (Lai et al., 2020; van Wessem et al., 2023). In the case of the disintegration of Larsen A and B ice shelves in 1995 and 2002,





respectively, depletion of the firn air content and subsequent meltwater ponding and infilling of crevasses is generally thought to have caused hydrofracturing(Scambos et al., 2000; Banwell et al., 2013). Surface meltwater accumulation on ice shelves can

trigger ice-shelf collapse when tensile stresses are sufficiently large (Banwell et al., 2013). Currently, 60 % of the ice shelves (by area) both buttress upstream ice and are vulnerable to hydrofracturing if experiencing accumulation of meltwater (Lai et al., 2020).

Perennial firn aquifers (PFAs), which are year-round subsurface bodies of liquid water within the firn's pore space, also modulate meltwater draining into crevasses at the bottom of the firn layer, potentially causing ice shelves to break up. PFAs

form in a climate with a combination of high surface melt rates and high snow accumulation rates (Kuipers Munneke et al., 2014). As snow has a low thermal conductivity (Calonne et al., 2019), high snowfall rates in fall and winter rapidly cover and insulate the summer meltwater and thereby prevent it from refreezing in winter.

PFAs are common across southeast Greenland (e.g., Forster et al., 2014; Miège et al., 2016). On the AIS, firn aquifers have been observed on the Wilkins and Müller ice shelves in the western Antarctic Peninsula (AP) (Montgomery et al., 2020;

MacDonell et al., 2021), the warmest and wettest region of Antarctica. An exploratory firn modelling study predicted PFAs to occur also on the grounded ice along the north-western coast of the AP (van Wessem et al., 2021). Combining regional climate model (RCM) with satellite data confirm a high probability of PFA occurrence in all these regions (Di Biase et al., 2024). Increasing surface melt and snowfall may result in future PFA expansion, also to other regions of Antarctica. Similarly, an inland expansion of aquifers occurred over the last decades in Greenland (Horlings et al., 2022).

While direct evidence linking PFAs to ice-shelf instability is currently lacking, Wilkins and Müller ice shelves, where firn aquifers have been observed, have both lost a considerable portion of their surface areas (33 and 49 %, respectively) since the 1950s (Cook and Vaughan, 2010). Neighbouring Jones and Wordie ice shelves, with similar climaticx conditions, have completely disintegrated (Cook and Vaughan, 2010). The absence of ice shelves along most of the north-western AP coast also suggests that the combination of ice shelves and firn aquifers is not viable.

PFAs can also play an important role over the grounded ice. In Greenland liquid water from PFAs drains through crevasses to the bed (Poinar et al., 2017), leading to basal lubrication, at least temporarily increasing ice velocity and ice discharge into the ocean (Zwally et al., 2002). Warming through the release of latent heat during refreezing changes the ice rheology (Hubbard et al., 2016). Hence, understanding the future evolution of PFAs is relevant when assessing the future stability of ice shelves and grounded ice.

Offline firn models forced by output of RCMs have proven to be useful tools to simulate the ice-sheet wide transient evolution of firn. They have already been used to simulate historical firn aquifers over Greenland (Brils et al., 2024), and the AP (van Wessem et al., 2021). RCMs are better suited for this than general circulation models (GCMs), whose resolution is too coarse to correctly represent strong coastal gradients in precipitation and surface melt (Bozkurt et al., 2021), which are particularly important for PFA formation (van Wessem et al., 2021). In addition, most GCMs do not properly represent important physical

processes, such as the snowmelt-albedo feedback (Jakobs et al., 2019).

For future projections of PFA extent, multiple climatic scenarios, and combinations of RCMs and GCMs are ideally used, as projections of near-surface climatic conditions over Antarctica vary widely among models (Carter et al., 2022; Kittel et al.,



2021). However, running a firn model for multiple forcings is computationally demanding. Therefore, we developed an XG-Boost PFA emulator, which is a machine learning model that mimics a more complex firn model. Emulators have previously been used to simulate firn (Dunmire et al., 2024; Verjans et al., 2021; Jourdain et al., 2024), as well as other ice sheet processes (Van Katwyk et al., 2023).

We train and evaluate the PFA emulator using output from the IMAU Firn Densification model v1.2AD (IMAU-FDM v1.2AD), which was recently updated by Veldhuijsen et al. (2024d) to better simulate transient firn densification in a changing climate. IMAU-FDM was forced by future climate realisations of the GCM CESM2, dynamically downscaled over the entire AIS by the RCM RACMO2.3p2 to a 27 km resolution, for emission scenarios SSP1-2.6, SSP2-4.5 and SSP5-8.5 (Veldhuijsen et al., 2024d). We subsequently apply our emulator to 12 climate projections from three polar RCMs (RACMO2.3p2, MAR3.11 and HIRHAM5) forced by several GCMs. We then report the resulting PFA distribution and expansion on the AIS.

## 2 Methods

### 2.1 IMAU-FDM firn model and forcing

IMAU-FDM version v1.2AD is a semi-empirical 1D firn densification model that simulates the time evolution of firn depth, density, temperature, grain size and liquid water content. Firn compaction is calculated based on the semi-empirical dry-snow densification equations of Arthern et al. (2010) with an updated dynamical densification expression to cope with changing climate forcing (Veldhuijsen et al., 2024d). The updated densification rate depends on firn temperature, grain size and overburden pressure instead of firn temperature and averages over the past 40 years of accumulation and surface temperature. The vertical percolation of liquid water from melt or rain is simulated using the bucket method, whereby liquid water is only retained through capillary forces (i.e. irreducible water). The maximum irreducible water decreases with increasing density (Coléou and Lesaffre, 1998). The meltwater can percolate through all layers in a single time step and (partly) refreeze when it reaches a layer with nonzero pore space and a temperature below the freezing point. Once the liquid water content of the lowermost firn layer exceeds the maximum irreducible water content, the surplus is assumed to instantaneously leave the firn column as runoff. The bucket method is computationally efficient but does not allow for saturated pore spaces, preferential flow, standing water over ice layers or horizontal flow.

In the simulations presented here, IMAU-FDM is forced at the upper boundary, the snow surface, with 3-hourly values of snowfall, sublimation, snowdrift erosion, 10-m wind speed, surface temperature, surface melt and rainfall from simulations of the RCM RACMO2.3p2 at a 27 km resolution (van Wessem et al., 2023). RACMO2.3p2 was in turn forced by the Community Earth System Model version 2 model output (CESM2) (Danabasoglu et al., 2020). These simulations consist of a historical climate realisation (1950–2014) and climate realisations for low-, middle- and high-emission future (2015–2100) scenarios (SSP1-2.6, SSP2-4.5 and SSP5-8.5, respectively). The projected Antarctic end-of-century (2090-2100) warming in SSP5-8.5 in CESM2 (+6.7 K) compared to 2005-2015 is stronger than the mean Antarctic warming in CMIP6 models (+4.8 K), which enables us to train the firn model on a wide range of temperatures (Dunmire et al., 2022; Kittel et al., 2021). The coupling between RACMO2.3p2 and IMAU-FDM is unidirectional. The stand-alone approach of IMAU-FDM allows for a



more realistic initialisation and higher vertical resolution than RACMO's built-in firn model, which uses similar physics. The disadvantage is that interaction with the atmosphere is not possible. For further details on the model set up and forcing, we refer to Veldhuijsen et al. (2024d).

## 2.2 PFA emulator

We develop a PFA emulator that mimics how IMAU-FDM transiently simulates PFAs. We use the yearly perennial amount of liquid water content (LWC) as the target variable, see next section. To emulate PFAs we develop an extreme gradient boosting (XGBoost) machine learning model (Chen and Guestrin, 2016). The model is developed in Python using the open-source scikit-learn and XGBoost packages. XGBoost sequentially builds an ensemble of weak decision trees. In each iteration, it fits a new tree to the residuals of the previous iteration, optimizing a specific objective function to minimize the overall prediction
error.

XGBoost is chosen for its high predictive accuracy and has been shown to outperform other machine learning algorithms, including neural networks, random forest regression and linear regression methods, in predicting e.g., snow water equivalent, snow avalanche susceptibility and glacier mass balance (Anilkumar et al., 2023; Iban and Bilgilioglu, 2023; Sun et al., 2024) and in general for medium and large sized (>10,000) tabular datasets (Grinsztajn et al., 2022). Additionally, XGBoost is highly
scalable, meaning it can handle large datasets and complex models, it incorporates regularization techniques to prevent overfitting and it has built-in mechanisms to estimate feature importance. Recently, a Random Forest model, a bagging algorithm that trains decision trees in parallel, has been used to emulate firn air content over Antarctica (Dunmire et al., 2024). While Random Forest is known for its simplicity, ease-of-use, and resistance to overfitting, XGBoost often better captures complex nonlinear relationships, especially in scenarios where there are interactions between features or when the dataset is high-dimensional
(Fatima et al., 2023).

### 2.2.1 Target variable and input features

Our target variable, the yearly perennial LWC, is defined as the minimum vertically integrated LWC over a year, based on model output at 10-day intervals. A yearly perennial LWC of zero indicates the absence of a PFA. A drawback of this approach is that it does not differentiate between years with brief periods without LWC (e.g. 10 days) and those with long periods without
LWC (e.g. 10 months). In addition, since LWC cannot be negative, the minimum value is zero, preventing negative values from counterbalancing positive biases. To resolve this, we introduce a negative value to represent the number of days without LWC. For instance, if LWC is absent for 10 days, the target value is -10; if LWC is absent for 10 months, the target value is about -300. Although the negative and positive quantities are very distinct, they are of the same order of magnitude, which provides a relatively smooth transition around zero.

The climate variables used as input features for our emulator are: (1) total annual snow accumulation (snowfall minus evaporation/sublimation), (2) total autumn (MAM) snow accumulation, (3) total summer (DJF) snowfall, (4) total annual liquid water input (surface melt plus rainfall), (5) annual melt-over-accumulation (MoA) ratio, (6) mean annual surface temperature, (7) mean summer (DJF) surface temperature and (8) annual seasonal temperature amplitude. These are the most important





mass fluxes and boundary conditions governing the firn density, temperature and LWC, and therewith the presence of PFAs

(Kuipers Munneke et al., 2014; van Wessem et al., 2021). For the summer, we consider snowfall instead of snow accumulation, due to the likely presence of evaporation alongside sublimation, which has different implications for PFAs. The annual seasonal temperature amplitude is defined as the difference between the average temperature of the warmest month and the coldest month. To account for the time it takes for firn to adjust to climatic conditions, we consider averages over the past 5, 10 and 30 years of the input features. Additionally, we include surface elevation and surface slope as input features, as the topography

influences short-term climate variability, such as extreme precipitation (González Herrero et al., 2023).

### 2.2.2 Dataset selection

The emulator is trained to predict yearly perennial LWC across Antarctica. The dataset consists of 18,136 IMAU-FDM grid cells with 85 time steps each (years between 2015-2100), amounting to 4.6 million data points for three climate scenarios. However, for most of the AIS, the climate conditions are not favorable for aquifer formation, i.e. too dry and/or too cold, even

under strong future warming. For instance, RACMO forced by CESM2 predicts that 76 % of the AIS does not experience any melt by the end of the century even in the SSP5-8.5 scenario (Veldhuijsen et al., 2024d). To minimize the influence of non-aquifer locations, we limit the training dataset to aquifer-favorable locations. For each scenario, we select data points where PFAs are present or where the duration without LWC is shorter than 150 days. In addition, we select data points with aquifer-favorable conditions, defined as having more than $400 \mathrm{~mm}\,\mathrm{w.e.}\,\mathrm{yr}^{-1}$ of accumulation and more than $200 \mathrm{~mm}\,\mathrm{w.e.}\,\mathrm{yr}^{-1}$ of

melt (van Wessem et al., 2021; Brils et al., 2024). From the remaining dataset, we randomly select 5,000 points from each scenario. In total, our dataset used for training consists of 123,000 data points (2.6 % of the total AIS data points).

### 2.2.3 Emulator training

Our dataset has a spatial and temporal structure, and to ensure independence between training and testing data, we split the training and testing data strategically rather than randomly (Roberts et al., 2017). Firstly, we evaluate the performance using

spatial blocking cross-validation. For this, we divide the area in which firn aquifers occur into eight regions. During the cross-validation, we leave out one region at a time. Since firn aquifers are rare on the AIS, cross-validation ensures that the model is evaluated on the full dataset. Additionally, we also evaluate the performance by leaving out each of the three emission scenarios at a time, referred to as scenario blocking cross-validation. The initial 30 years of the future part of the simulations (2015-2045) are excluded from the validation scores to avoid dependency on the other scenarios. Finally, we use all scenarios and regions

for training the emulator, to ensure that the widest possible range of warming trends and climatic conditions are included in the emulator.

The hyperparameters are tuned using the BayesSearchCV function (built-in scikit-learn) during the spatial blocking cross-validation. Eight key hyperparameters involved in the XGBoost algorithm were optimized. Firstly, the number of estimators (`n_estimators`) refers to the number of trees. The learning rate (`learning_rate`) controls the step size at which the

optimizer makes updates to the weights. The regularisation techniques included in XGBoost are the maximum depth of the trees (`max_depth`), the minimum weight needed in a leaf node (`min_child _weight`), the subsample ratio of the training





**Table 1.** List of available forcing datasets of regional climate models (RCMs) driven by global climate models (GCMs) along with the GCM era, GCM ensemble member number, RCM horizontal resolution and the available scenarios. We use the period 1980-2100 of all forcing datasets.

| RCM | GCM | GCM era | GCM member | RCM horizontal resolution | Scenarios |
|---|---|---|---|---|---|
| RACMO | CESM2 | CMIP6 | r11i1p1f1 | 27 km | SSP1-2.6, SSP2-4.5, SSP5-8.5 |
| MAR | ACCESS-1.3 | CMIP5 | r1i1p1 | 35 km | SSP5-8.5 |
| MAR | CESM2 | CMIP6 | r11i1p1f1 | 35 km | SSP1-2.6, SSP2-4.5, SSP5-8.5 |
| MAR | NorESM1-M | CMIP5 | r1i1p1 | 35 km | SSP5-8.5 |
| MAR | CNRM-CM6-1 | CMIP6 | r1i1p1f2 | 35 km | SSP5-8.5 |
| HIRHAM | CESM2 | CMIP6 | r11i1p1f1 | 12 km | SSP1-2.6, SSP5-8.5 |
| HIRHAM | EC-Earth3 | CMIP6 | r5i1p1f1 | 12 km | SSP5-8.5 |

instances (`subsample`), subsample ratio of columns when constructing each tree (`colsample_bytree`), the minimum loss reduction (`gamma`), and the regularisation degree (`reg_lambda`).

### 2.3 Predicting firn aquifers

IMAU-FDM simulations forced by RACMO-CESM for three scenarios (SSP1-2.6, SSP2-4.5 and SSP5-8.5) were used to train the PFA emulator. By applying the emulator to output from three polar RCMs, an ensemble of 12 scenarios of perennial LWC predictions for 2015-2100 is produced (Table 1). CESM2-forced runs of MAR (Jourdain et al., 2024) and HIRHAM (Hansen et al., 2021) RCMs are available for SSP1-2.6, SSP2-4.5 and SSP5-8.5, and SSP1-2.6 and SSP5-8.5, respectively. In addition, MAR simulations driven by GCMs ACCESS-1.3, NorESM1-M and CNRM-CM6-1 for SSP5-8.5 (Kittel et al., 2021)

and HIRHAM simulations driven by GCMs EC-Earth3 for SSP5-8.5 (Boberg et al., 2022) were also used. The horizontal resolution of each climate forcing is preserved in the emulator simulations.

Even for the contemporary AIS climate, significant differences in near-surface air temperature, snowfall and melt exist between these three RCMs (Mottram et al., 2021; Carter et al., 2022). An important difference between the RCMs lies in the handling of precipitation advection: MAR allows precipitation to be advected through the atmospheric layers until reaching

the surface, whereas RACMO and HIRHAM deposit precipitation instantaneously. Furthermore, RACMO and MAR use subsurface schemes optimized over snow and ice for Antarctica, while HIRHAM applies simpler surface snow physics for ice surfaces. In addition, notable climatic differences between the GCM simulations exists; for instance, the 21st-century AIS warming for the four selected GCMs for MAR for SSP5-8.5 ranges from +3.2 K (NorESM1-M) to +8.5 K (CNRM-CM6-1) (Kittel et al., 2021).





## 3 Emulator tuning and evaluation

### 3.1 Emulator tuning

The eight regions used for the spatial blocking cross-validation are shown in Fig. 1. The optimal hyperparameters obtained during the spatial blocking cross-validation are listed in Table 1. We found that removing the 30-year average melt, rain and surface temperature input features improved the performance. Therefore, we leave out the 30-year melt input feature. The 30-year surface temperatures were replaced by the 30-year averaged temperature trend. This led to a small improvement in the performance (3 % less false negatives and 1 % less false positives).

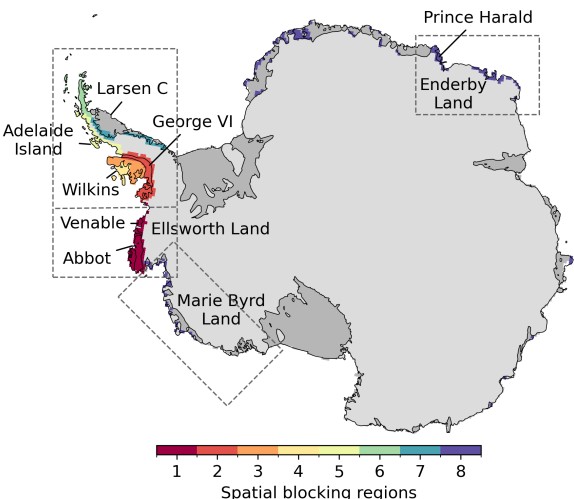

**Figure 1.** Regions used for spatial blocking cross-validation. The names indicate locations referred to in the text and the dashed boxes indicate the regions in Figs. 3, 5, 7, 8, 10, S1 and S4.

### 3.2 Emulator evaluation

During the spatial blocking cross-validation, the emulator explains 89 % of the variance in the IMAU-FDM perennial LWC ($R^2 = 0.90$; Fig. 2a). The RMSE is 86 mm or days, and the bias is 0.4 mm or days. However, in this work we aim to predict the presence or absence of aquifers rather than perennial LWC. When employing a positive perennial LWC threshold to identify aquifers, the emulator successfully predicts 88 % of the IMAU-FDM aquifers (12 % false negatives) with 9 % false positives. The emulator yields an $R^2$ value of 0.90 for the years with PFAs of the individual locations (Fig. 2c). Maps of PFA years of the AP and Ellsworth Land for the spatial blocking for SSP5-8.5 are shown in Fig. 3. The performance is poorest on and around the Larsen C and D ice shelves (region 7, RMSE = 8 years), with under- and overestimates. When randomly instead of strategically splitting our training and testing data, the emulator yields a misleadingly high $R^2$ value of 0.98 which highlights the importance of using strategically splitting the data to prevent overfitting.



**Table 2.** Hyperparameter ranges used for tuning and their optimal values.

| Parameter | Tuning range | Optimal value |
| --- | --- | --- |
| n_estimators | 0-300 | 179 |
| learning_rate | 0.001-1 | 0.034 |
| min_child_weight | 0-50 | 44 |
| max_depth | 2-15 | 6 |
| subsample | 0.5-1 | 0.58 |
| colsample_bytree | 0.5-1 | 0.77 |
| gamma | 0-10 | 4.2 |
| reg_lambda | 0-10 | 2.8 |

During the validation in which SSP2-4.5 is left out, the emulator yields an $R^2$ value of 0.96 for both the perennial LWC and the years with PFAs (Figs. 2b,d). For this, we include all data of all locations where either the emulator or the model forecasts the presence of an aquifer or predicts fewer than 150 days without LWC in at least one year. The emulator successfully predicts

88 % of the IMAU-FDM aquifer years of the SSP2-4.5 scenario (12 % false negatives), with 7 % false positives. The results for leaving out SSP1-2.6 are similar, with the emulator predicting 92 % of the aquifer years with 11 % false positives ($R^2$ = 0.96 and 0.97 for perennial LWC and years with PFAs, respectively). When leaving out the SSP5-8.5 scenario, the emulator predicts 89 % of the aquifer years, with 15 % false positives ($R^2$ = 0.9 and 0.91 for perennial LWC and years with PFAs, respectively). The larger error can be attributed to the large differences in warming trend.

Four example time series from the spatial blocking and scenario blocking evaluations are shown in Fig. 4. These figures show that the emulator can mimic the growth as well as the shrinking of aquifers. They also illustrate that deviations from the firn model mainly occur during the onset of aquifers, when aquifers are relatively small.

## 4 Results

### 4.1 PFAs predictions under scenarios SSP1-2.6 and SSP2-4.5

The trained emulator predicts PFAs over the period 2015-2100 for the 12 forcings listed in Table 1. It should be noted that all available SSP1-2.6 and SSP2-4.5 RCM forcings are from CESM2. For SSP1-2.6 and SSP2-4.5, PFAs are projected to remain mostly restricted to the AP, except for expansion to Enderby Land (East Antarctica) under SSP2-4.5 for MAR-CESM (Fig. 5). All forcings predict PFAs along the north-western coast of the AP. Additionally, RACMO-CESM forcings predict PFAs on Wilkins ice shelf for SSP1-2.6 and SSP2-4.5, and on George VI ice shelves for SSP2-4.5. MAR-CESM forcing also predicts

aquifers on Wilkins and George VI ice shelves for SSP2-4.5, and both MAR forcings predict aquifers along the grounding lines of Larsen C ice shelf. Figure 6a shows time series of PFA extent for all SSP1-2.6 and SSP2-4.5 simulations. As all RCMs are



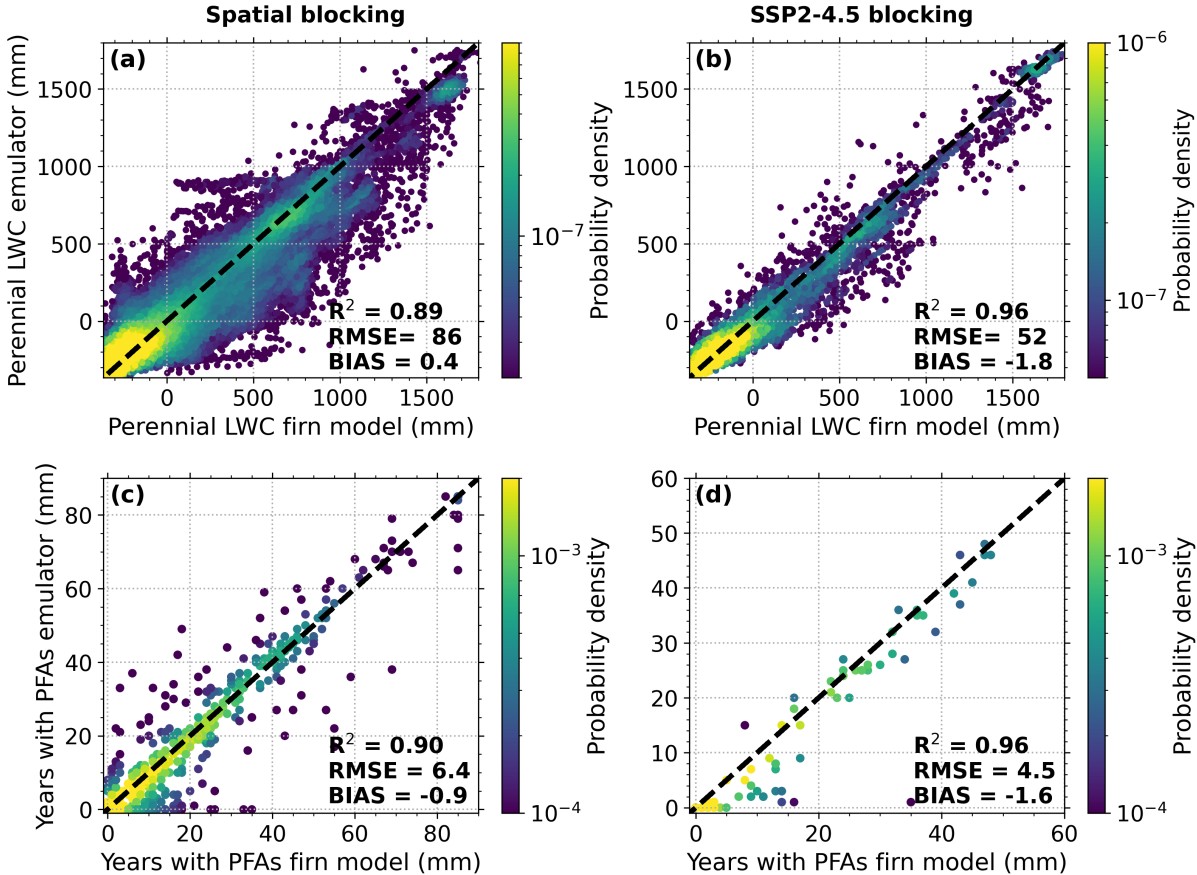

**Figure 2.** **(a,b)** Perennial liquid water content (LWC) and **(c,d)** number of years with perennial firn aquifers (PFAs) from the **(a,b)** spatial blocking cross-validation and **(c,d)** SSP2-4.5 block validation. Panels **(a)** and **(b)** include the data points used for training and evaluation described in Sect. 2.2.2. Panels **(c)** and **(d)** include all locations where either the emulator or the model forecasts the presence of an aquifer or predicts fewer than 150 days without LWC in at least one year in the associated scenario.

forced with the same two CESM2 realisations, peaks and minima in modelled PFA extent align. MAR predicts the highest PFA extent by 2100 for SSP1-2.6 ($50,000\,\mathrm{km}^2$) and SSP2-4.5 ($87,000\,\mathrm{km}^2$), followed by RACMO ($31,000$ and $43,000\,\mathrm{km}^2$), while HIRHAM-CESM only predicts a PFA extent of $19,000\,\mathrm{km}^2$ for SSP1-2.6. Part of the difference between the RCMs arises by the distinct horizontal resolutions. Overall, PFA expansion starts to accelerate around 2050 in these runs.

**4.2 PFAs predictions under scenario SSP5-8.5**

For SSP5-8.5, PFAs expand within the AP, and to Ellsworth Land, on and around Abbot and Venable ice shelves, except for MAR-NorESM, which only expands within the AP (Fig. 7). All forcings predict aquifers along the north-western coast of the AP, and along the grounding lines of Larsen C, George VI and Wilkins ice shelves. In addition, RACMO-CESM stands out, as



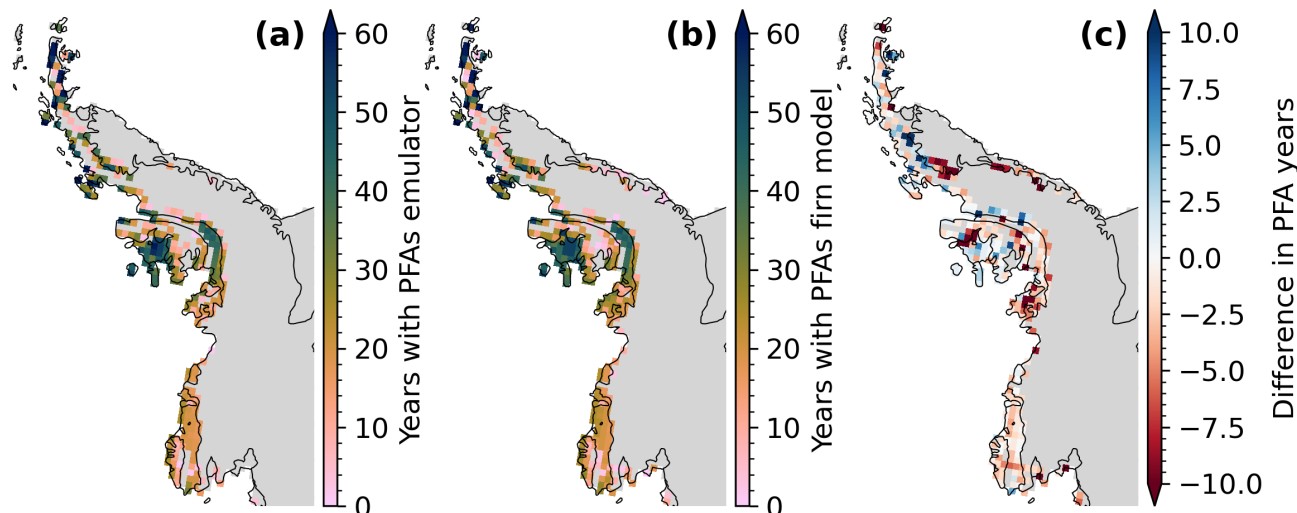

**Figure 3. (a)** Number of years with perennial firn aquifers (PFAs) under SSP5-8.5 from the emulator during the spatial blocking validation and **(b)** from the IMAU-FDM firn model and **(c)** the difference.

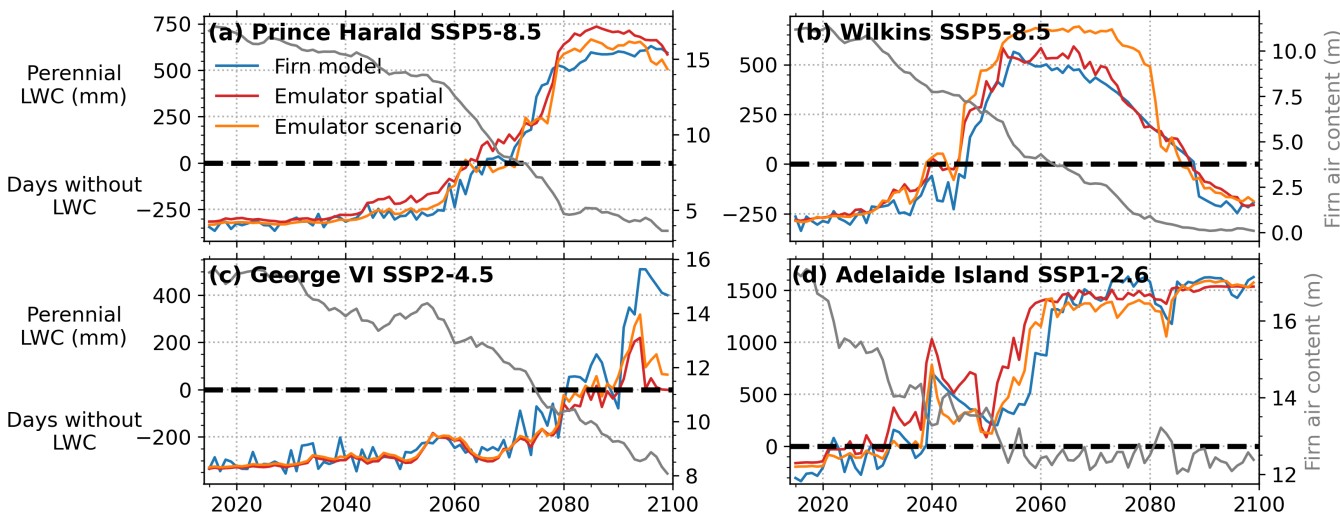

**Figure 4.** Example time series of perennial liquid water content (LWC)/days without LWC for the firn model (blue) and the emulator for spatial blocking (red) and for scenario blocking (orange), as well as firn air content for the firn model (gray), for individual grid points and different scenarios. **(a)** and **(b)** are SSP5-8.5, **(c)** is SSP2-4.5, and **(d)** is SSP1.2-6. The locations of the grid points are indicated in Fig. 1.

it predicts extensive PFA coverage on Wilkins, George VI and Abbot ice shelves (between 70-100 % of the ice shelf area). In contrast, all MAR forcings and HIRHAM-EC-Earth predict extensive PFA coverage on Larsen C ice shelf (>12,000 $km^2$). PFAs are also predicted by 5 out of the 7 models in Enderby Land in East Antarctica (Fig. 8a). MAR-ACCESS and MAR-NorESM





**Figure 5.** Number of years with perennial firn aquifers (PFAs) for **(a)** SSP1-2.6 and **(b)** SSP2-4.5 forcing datasets over the period 2015-2100 for the Antarctic Peninsula. The MAR-CESM SSP2-4.5 forcing also includes Enderby Land.

are the two models that do not predict PFAs in this region, which also predict lowest overall PFAs extent (100,000 $km^2$ and 127,000 $km^2$ by 2100, respectively, Fig. 6b). For HIRHAM-EC-Earth, and MAR-CNRM-CM6, the emulator predicts PFAs

in Marie Byrd Land. MAR-CNRM predicts the largest PFA extent by 2100, 334,000 $km^2$, followed by HIRHAM-Ec-Earth, 289,000 $km^2$. Notably, HIRHAM-EC-Earth predicts at least twice the initial (2015) PFA extent (72,000 $km^2$) compared to the other simulations. Using the same GCM forcing (CESM2), RACMO and MAR yield comparable PFA extent (257,000 and 234,000 $km^2$), while HIRHAM-CESM predicts a considerably lower extent (148,000 $km^2$). On the other hand, for SSP1-2.6





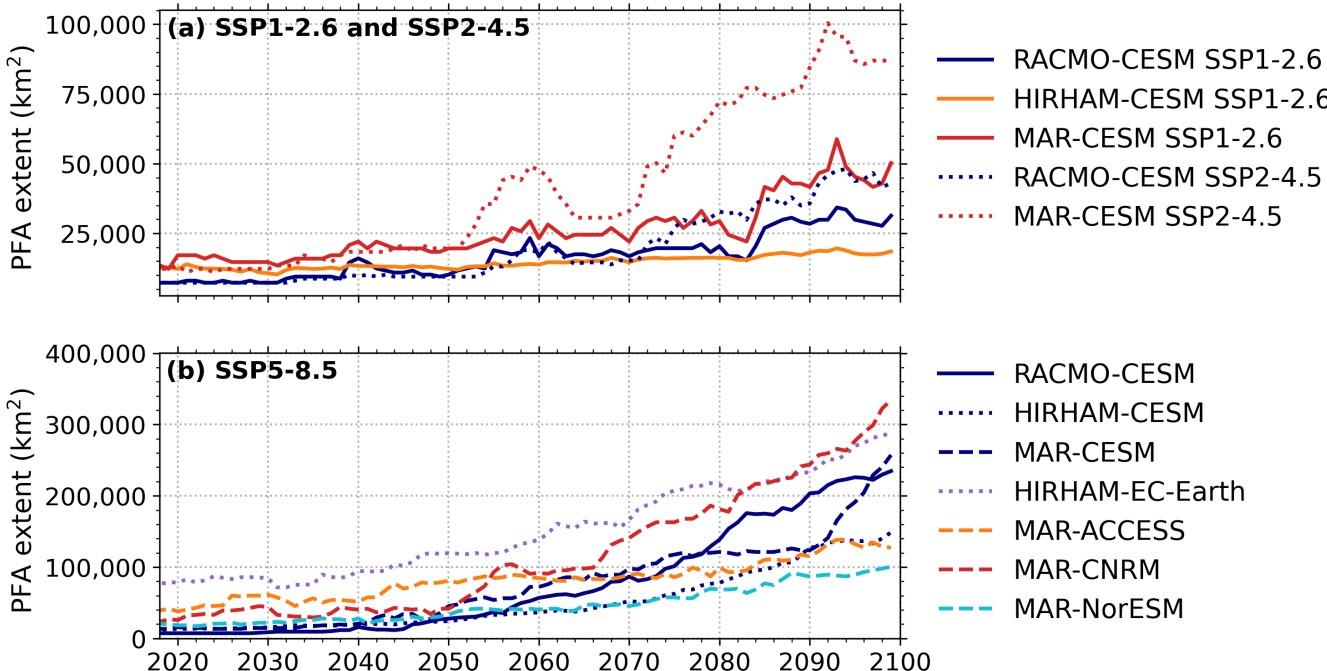

**Figure 6.** Time series of perennial firn aquifer (PFA) extent for all climatic forcing datasets for **(a)** SSP1-2.6 and SSP2-4.5 and **(b)** SSP5-8.5 scenarios.

and SSP2-4.5, MAR yields higher PFA compared to RACMO for the same GCM forcing. PFA expansion starts to accelerate

around 2050 for the CESM2 and CNRM-CM6 forcings, similarly as for SSP1-2.6 and SSP2-4.5 (Fig. 6).

### 4.3 Climatic drivers

The previous sections show a large range of PFA predictions, caused by differences in the climatic forcing. Firstly, differences in initial surface temperature and warming between the forcing datasets explain part of the spread (Fig. 9b). For example, MAR-NorESM predicts the smallest PFA extent for SSP5-8.5 by 2100, as it has the smallest AIS end-of-century (2090-2100)

warming (+3.3 K) compared to 2005-2015, and lowest end-of-century surface temperature (241.6 K). On the other hand, HIRHAM-EC-Earth and MAR-CNRM-CM have the highest temperatures by 2090-2100 for SSP5-8.5 (245.9 and 244.9 K, respectively) with a warming of +4.6 and +6.6 K, respectively, and predict the largest PFA extent. For HIRHAM-EC-Earth, the initial temperature is high, related to the warm bias over Antarctica in EC-Earth3 (Boberg et al., 2022), which explains the high PFA extent at the start of the simulation. For SSP1-2.6 and SSP2-4.5, the largest PFA extent is predicted by MAR-CESM,

which is also the warmest model combination (Fig. 9a).

Significant regional differences also exist between RCMs for the same GCM forcing, in this case CESM2. For example, on Larsen C ice shelf most aquifers are predicted by MAR, which is related to high surface melt and high accumulation at the foot





**Figure 7.** Number of years with perennial firn aquifers (PFAs) for SSP5-8.5 forcing datasets for the Antarctic Peninsula and Ellsworth Land.

of the mountains, which are absent in the other two models (Fig. 10; see Fig. S2 for surface temperature). On the other hand, the absence of extensive PFAs on Wilkins and George VI ice shelves in MAR and HIRHAM is due to lower accumulation compared to RACMO. Furthermore, HIRHAM-CESM calculates lower surface melt rates than the other two models, despite mean surface temperatures in HIRHAM-CESM being comparable to those in RACMO-CESM.

Figure 11a shows that the presence of PFAs for RACMO and MAR is strongly governed by the rates of accumulation and melt (plus rain), here shown as averages for the preceding 10 years. It shows that at least $300 \, \mathrm{mm \, w.e. \, yr^{-1}}$ of surface melt and



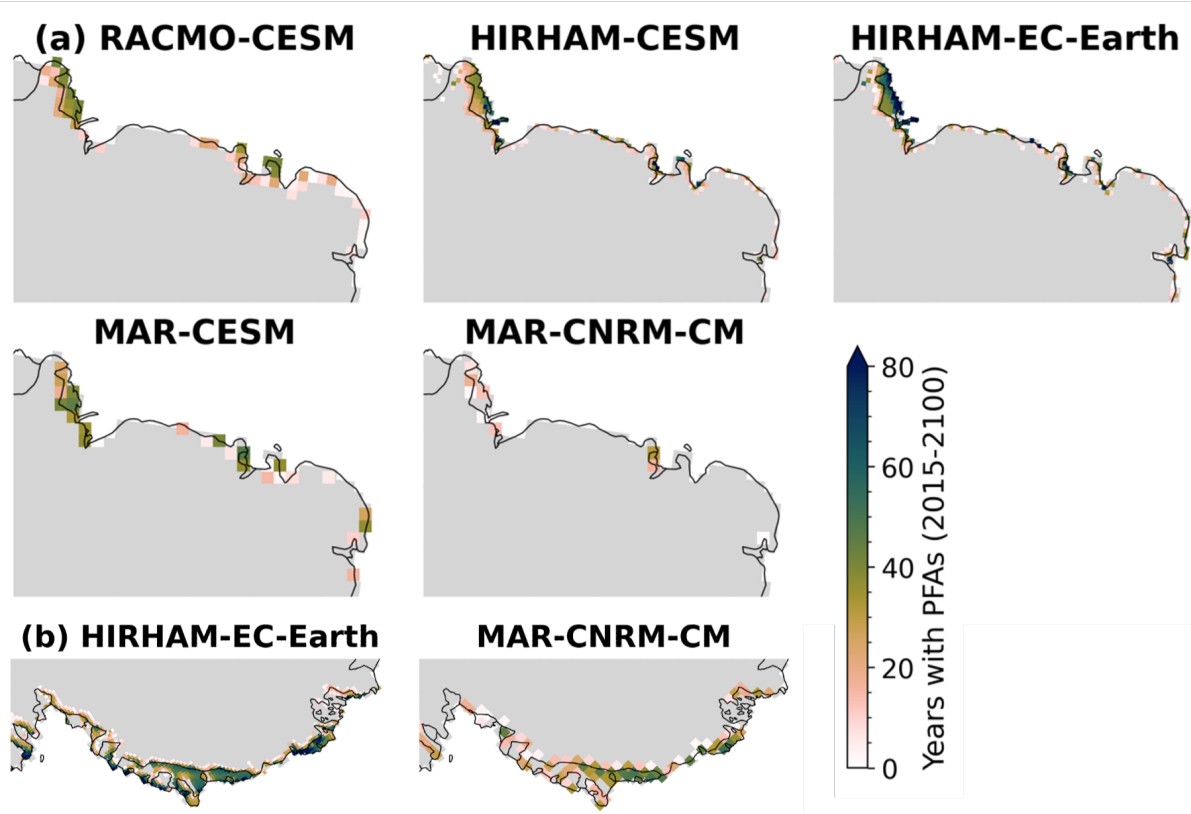

**Figure 8.** Number of years with perennial firn aquifers (PFAs) for SSP5-8.5 forcing datasets for **(a)** Enderby Land and **(b)** Marie Byrd Land.

600 mm w.e. yr$^{-1}$ of accumulation are required for aquifers to form. When melt and accumulation exceed 600 mm w.e. yr$^{-1}$
and 1000 mm w.e. yr$^{-1}$, respectively, an aquifer is predicted in nearly all cases, in line with Brils et al. (2024). The same figure
for HIRHAM (Fig. S2a) suggests that PFAs also form at slightly lower surface melt and accumulation rates. The is because
HIRHAM models considerably less surface melt and accumulation for a given temperature than MAR and RACMO (Fig. S3).
The results of the emulator for HIRHAM are thus less reliable, as its combinations of temperature, accumulation and surface
melt rates are not present in the training data. Therefore, in the remainder we focus on the emulator results from RACMO and
MAR forcings.

## 4.4 Transient PFAs

As Figure 4b shows, PFAs can also develop, shrink and subsequently disappear, henceforth referred to as transient PFAs. A
transient PFA is defined as a PFA that existed for at least 5 years but has disappeared before 2100. Figure 11b shows the presence
of transient PFAs as a function of melt (plus rain) against accumulation rates for RACMO and MAR simulations. The white
dotted lines indicate the melt-over-accumulation (MoA) ratios of 0.7 and 1.7, which are theoretical thresholds for indicating





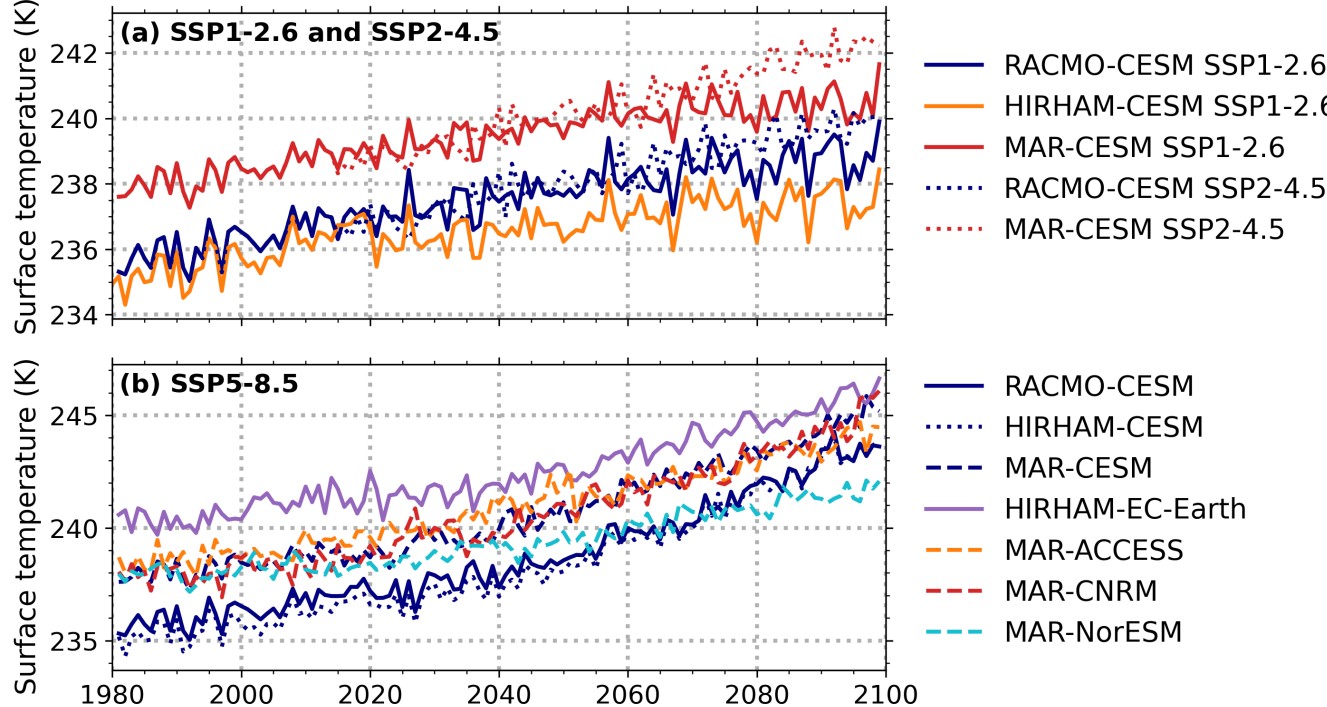

**Figure 9.** Time series of annual average AIS surface temperature for all climatic forcing datasets for **(a)** SSP1-2.6 and SSP2-4.5 and **(b)** SSP5-8.5 scenarios.

ice slabs and ablation zone, respectively (Brils et al., 2024). Transient PFAs mainly occur once the 0.7 and especially the 1.7 threshold have been exceeded, and mainly below accumulation rates of 1000 mm w.e. yr$^{-1}$. Figure S4 shows the locations of these transient PFAs for the RACMO and MAR SSP5-8.5 simulations. Transient PFAs mainly occur along the boundaries of PFA regions. Most notably, transient PFAs occur on Wilkins ice shelf in RACMO-CESM for the SSP5-8.5 scenarios. Transient

PFAs mainly occur under SSP5-8.5, covering on average 28,000 km$^2$ by 2100, which is 13 % compared to the 2100 PFA extent (Fig. 11). In the SSP1-2.6 and SSP2-4.5 scenarios the extent of transient PFAs is on average only 4,200 km$^2$, 9 % of the 2100 PFA extent.

## 5 Discussion

### 5.1 Firn model and observations

Since our emulator is trained on IMAU-FDM data, the uncertainties inherent to the firn model will also persist in the emulator. The bucket method does not include preferential flow, which can lead to an underestimation of the percolation depth of liquid water and therefore aquifer recharge as observed in Greenland (Miller et al., 2018; Vandecrux et al., 2020). The firn model also




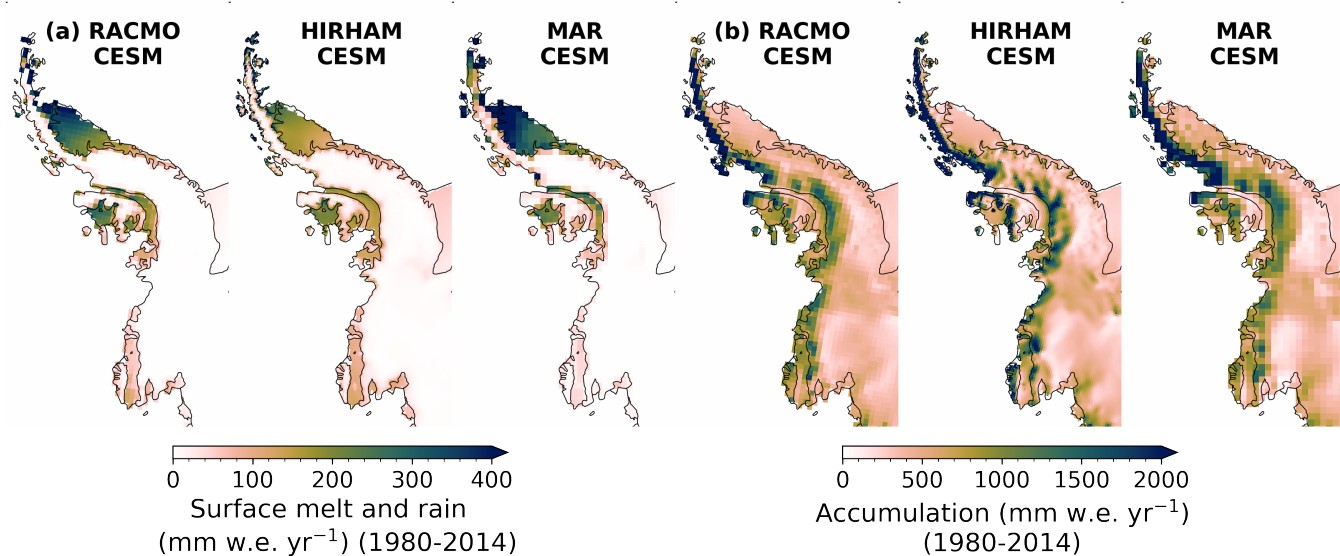

**Figure 10.** Annual average **(a)** surface melt and rainfall, and **(b)** snow accumulation over the historical period (1980-2014) for RACMO-CESM, HIRHAM-CESM and MAR-CESM.

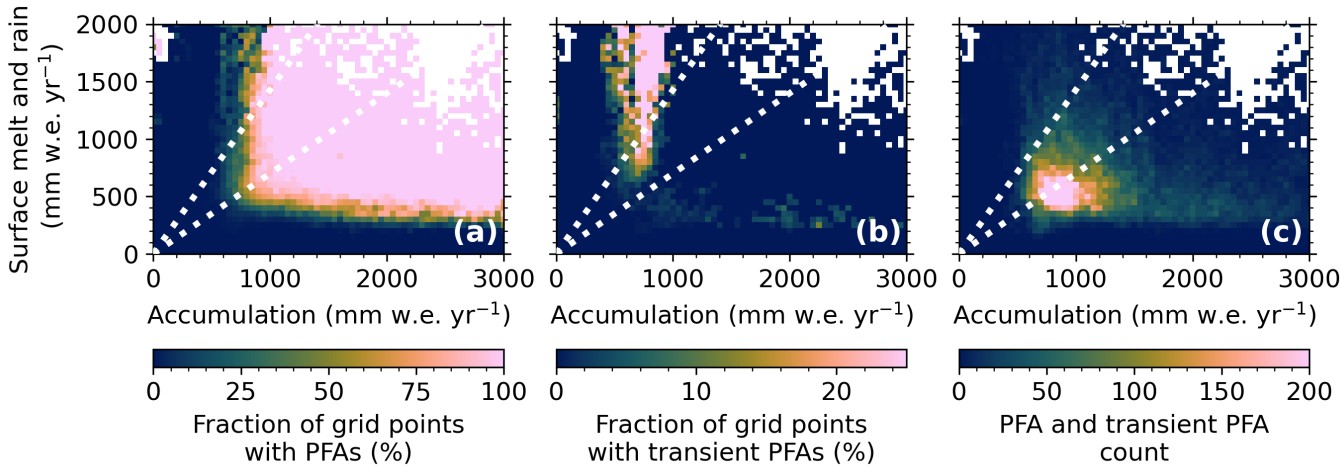

**Figure 11.** Percentage of **(a)** perennial firn aquifer (PFA) occurence and **(b)** transient PFA occurrence, and **(c)** total amount of PFA and transient PFA occurrence, from all RACMO and MAR simulations, as a function of annual surface melt and rainfall (y-axis) and snow accumulation (x-axis). The grid cells are grouped in melt and accumulation bins of $50 \, \text{mm w.e. yr}^{-1}$. The dotted lines indicate the 0.7 and 1.7 melt-over-accumulation (MoA) thresholds.

does not allow for lateral meltwater transport, which has been observed within aquifers, but with a limited measured specific discharge ($60 - 140 \, \text{m yr}^{-1}$) (Montgomery et al., 2020; Miller et al., 2018). In addition, the use of irreducible water content



hinders the ability to estimate the volume of meltwater stored within a PFA. Nonetheless, the bucket method is used as it is a fast and stable meltwater scheme, in contrast to many Richards-equation based models that are computationally more expensive and prone to numerical instabilities. While it is difficult to estimate the volume of meltwater stored within a PFA with the bucket method, the presence of year-round LWC in the firn can be used as an indication of PFA presence (Kuipers Munneke et al., 2014). Brils et al. (2024) simulated aquifers in Greenland using IMAU-FDM, agreeing with 62 % of the observed aquifers by airborne and ground-penetrating radar measurements. The mismatch can be explained by underestimation of detected aquifers by widely spaced radar data, resolution limitations of the firn model, drainage to crevasses that is not included in the model, in addition to the above described firn model limitations, and biases in the climatic forcing.

In Antarctica, PFAs have thus far only been observed in-situ on the Wilkins and Müller ice shelves (Montgomery et al., 2020; MacDonell et al., 2021). Using an integrated approach combining satellite and RCM data, PFAs are also predicted along the north-western coast of the AP, the warmest and wettest region of Antarctica, and on George VI ice shelf (Di Biase et al., 2024). PFAs along the north-western coast, indicated by the satellite as well as the RCM data, agree with our emulator results. However, historical PFA extent on Wilkins and George VI ice shelves appears underestimated in the emulator simulations. In contrast, on Larsen C ice shelf current PFAs are simulated by all MAR forcing datasets, while PFA probability is low here (Di Biase et al., 2024). Müller ice shelf is located in a region with complex topography, which is not resolved in the RCMs. Overall, the modelled historical PFA extent shows moderate agreement with observations, especially for intermediate accumulation and surface melt rates. As future warming leads to increased melt and accumulation, the emulator is expected to produce more accurate PFA predictions, as more locations will shift into distinct PFA climate regimes rather than remaining in transitional states.

## 5.2 XGBoost emulator

Our PFA emulator captures at least 89 % of the simulated PFA variance. Despite its demonstrated generalisation capabilities, we emphasize that XGBoost models can be prone to errors when used beyond their range of training conditions (Chen and Guestrin, 2016). Given the large spatial variability in climatic conditions across the AIS, the range of the individual input features are likely covered in the training data. However, our findings suggest that some specific combinations of these features may be absent in the training data. This is illustrated by the lower melt and accumulation for a given temperature in HIRHAM compared to RACMO and MAR. As a result, the emulator predicts PFAs for HIRHAM forcings under unrealistically low melt and accumulation rates (Brils et al., 2024; van Wessem et al., 2021). RACMO and MAR, are in better agreement on the conditions of PFA formation, reinforcing our confidence in the emulator's performance. However, if a different RCM is used for either training or predicting, the consistency between the RCMs should be evaluated to ensure reliable results. Additionally, our work highlights the importance of using spatial and scenario blocking instead of randomly splitting training and testing data. In the latter approach, the training and testing data are very similar, while with spatial or scenario blocking, XGBoost is trained and tested on datasets with slightly different characteristics. As a result, overfitting and overoptimistic perception of the predictive power is prevented (Roberts et al., 2017).



## 5.3 PFA expansion uncertainties

Overall, PFA expansion responds primarily to the modelled warming, either by a higher emission scenario, or by a warmer
model. The Larsen C ice shelf stands out in this regard, as all simulations by MAR clearly predict PFAs here, also in colder
runs, such as MAR-norESM (Figs. 5, 7). Generally, PFAs are predicted on the windward side of Antarctic and Greenland
mountain ranges, and receive moist and mild air masses originating from the sea, resulting favorable conditions for PFA
formation (Turner et al., 2019). The Larsen C ice shelf is considered too dry, as it is located on the leeward side of the AP
mountain range. This ice shelf is frequently affected by föhn winds, resulting in dry and warm conditions (Luckman et al.,
2014). Nevertheless, substantial PFAs are predicted here for MAR, as this is the only RCM that allows precipitation to be
advected over the mountains before reaching the surface. Advection of precipitation is also included in the recently developed
RACMO2.4 version (Van Dalum et al. 2024). However, as described in Section 5.1, MAR overestimates current PFA extent
on Larsen C ice shelf, suggesting that advection of precipitation might be overestimated. In addition, the coarse resolution of
MAR (35 km), contributes to this effect, as blocking of humid air is underestimated (Datta et al., 2018). This in turn, can cause
overestimation of clouds, precipitation, longwave radiation and consequently melt on Larsen C ice shelf.

The horizontal resolutions of all forcing datasets and thus emulator results ranges from 12 to 35 km. However, Van Wessem
et al. (2016) suggest that even at 5.5 km resolution, the underestimation of the height and slope of the orographic barrier may
result in an underestimation of precipitation and föhn winds in the AP. Another uncertainty concerning the emulator results is
that all SSP1-2.6 and SSP2-4.5 simulations are indirectly driven by CESM2, which has a high climate sensitivity, but also a
cold bias over the AIS (Dunmire et al., 2022).

## 5.4 Implication for future ice-shelf stability

As new PFAs are developing, the firn air content is also decreasing (Fig. 4). If warming continues, the firn air content eventually
becomes fully depleted, after which the PFAs disappear, making them transient. In fact, explicit firn simulations for Greenland
suggest that PFAs are a committed transition state from healthy firn to depleted firn in high accumulation regions (>1,000
$mm\,yr^{-1}$) (Brils et al., 2024). The duration of such transient PFA presence will decrease with stronger warming rates or lower
accumulation rates. Thus, the importance of including PFAs when assessing the timing of ice-shelf vulnerability also decreases.
This is illustrated by the stable PFA on Adelaide island (Fig. 4d), which has a high accumulation rate over the historical
period (3100 $mm\,yr^{-1}$) and low warming rate over the 21st century (+3.2 K). In contrast, on Wilkins ice shelf, the historical
accumulation rate is relatively low (800 $mm\,yr^{-1}$), and the warming rate is high (+7.2 K), resulting in the disappearance of
the PFA after 40 years (Fig. 4b). When combining our results with those of Lai et al. (2020), we find that Wilkins, Larsen C,
and Abbot ice shelves, where both PFAs are predicted and substantial tensile stresses occur, may be vulnerable to instability
due to PFAs.



## 6 Conclusions

An XGBoost machine learning emulator was developed to predict future Antarctic PFAs (2015-2100) for an ensemble of
12 scenarios from three RCMs (RACMO, MAR and HIRHAM) in combination with five GCMs. The emulator was trained
with simulations of three scenarios (SSP1-2.6, SSP2-4.5 and SSP5-8.5) from IMAU-FDM forced by RACMO-CESM. Using
a scenario and spatial blocking evaluation approach, we found that the emulator successfully explains at least 89 % of the
PFA presence and perennial LWC variance. Under SSP1-2.6 and SSP2-4.5, PFA presence remains restricted to the AP. For
SSP5-8.5, PFAs expand within the AP, and expand to Ellsworth Land in six out of the seven simulations, to Enderby Land
in East Antarctica in five out of the seven simulations and to Marie Byrd Land in two out of the seven simulations. There
is a large spread among the RCMs and GCMs predictions, related to differences in the climatic input, which highlights the
usefulness of the emulator. For climatic forcings from RACMO and MAR, we find that liquid water input (melt and rain)
and snow accumulation are good predictors for the occurrence of PFAs. However, HIRHAM predicts considerably lower melt
and accumulation for a given temperature compared to MAR and RACMO, causing less realistic PFA predictions. Overall,
our results show that, irrespective of the emission scenario, firn aquifers are likely to expand in a warmer Antarctica. This
highlights the importance to understand the impacts PFAs have on ice sheet hydrology, instability and ice-shelf stability.

*Code and data availability.* The code of IMAU-FDM v1.2AD is available at: https://doi.org/10.5281/zenodo.10723570 (Veldhuijsen et al.,
2023). The code of the XGBoost model is published at: https://zenodo.org/records/13750024 (Veldhuijsen et al., 2024a) The emulator results
(perennial liquid water content) are published at: https://zenodo.org/records/13626884 (Veldhuijsen et al., 2024b). The firn model results
(perennial liquid water content) are published at: https://doi.org/10.5281/zenodo.10723570 (Veldhuijsen et al., 2024c).

*Author contributions.* SbmV and WJvdB defined the research goals and designed the study. SV developed the XGBoost model, performed
the simulations and analyzed the results. NH and FB provided HIRHAM data, and CK and CA provided MAR data. All authors contributed
to discussions on the manuscript.

*Competing interests.* MRvdB is a member of the editorial board of journal The Cryosphere.

*Acknowledgements.* This work was funded by the Netherlands Organization for Scientific Research (grant no. OCENW.GROOT.2019.091).
FBO and NH is supported by the Danish State through the National Centre for Climate Research (NCKF), further NH is also supported by
the Novo Nordisk Foundation project PRECISE (NNF23OC0081251). We thank Kamilla Hauknes Sjursen for her valuable feedback on the
development procedure of the XGBoost model. MvdB acknowledges support from NESSC. SbmV used ChatGPT to enhance the readability
of specific sections. Following this, the content was reviewed and edited as necessary and full responsibility is taken for the final publication.



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
