# Peer review of "Emulating the expansion of Antarctic perennial firm aquifers in the $21^{st}$ century"

_EGUsphere, 2024_

## Referee Comment (RC1)

Review of **Emulating the future distribution of perennial firn aquifers in Antarctica**
Veldhuijsen et al. 2024

In this work, the authors develop an emulator that mimics the IMAU-FDM firn model to predict firn aquifer presence in Antarctica. They apply their emulator to an ensemble of different RCM-GCM pairs to evaluate how firn aquifer presence may change in different future emission scenarios. Most of the text and figures throughout the manuscript are clear; however, there are several important methodological details that are not well-explained. I have several major comments mainly related to the methodology. Because there were several things I did not fully understand, I found it difficult to assess the results and discussion. I believe this is an interested and exciting topic and support publication if these concerns are addressed.

**Major comments**

First, L80 reads: "The bucket method is computationally efficient but does not allow for saturated pore spaces, preferential flow, standing water over ice layers, or horizontal flow". Since a firn aquifer is quite literally "saturated pore space", I think a more in-depth explanation about why the bucket scheme can be used in this work is required.

The paragraph in lines 115-119 is very confusing. I don't understand the reasoning for why it is necessary to introduce a negative value to counterbalance the positive value. Also how are these values 'of the same order of magnitude'? In Figure 2a,b there are many points with LWC above 1000; however I would think the minimum possible value for the number of days without LWC is -365. When you say "we introduce a negative value…", what is meant by this? Do you average this value with the perennial LWC? Or is your model then predicting 2 separate target variables?

Also, why can you not just predict the average LWC during the winter months, instead of the annual minimum? As winter is the least-likely month for liquid water to exist, maybe this would be a simpler way to predict PFA presence?

Finally, from the target variable, what do you actually consider to be a PFA? This is unclear from the text.

Further, in section 4 (L184 ): It does not make sense to me to report the RMSE as 86 mm or days. I think I just don't understand the target variable but I believe it to be some combination of LWC and days without PFA presence it which case the RMSE would not strictly be 86mm or 86 days but some combination of the two?

Finally, a look into feature importance would be a very valuable addition to this work. Are there features that are more heavily utilized by the model than others for predicting LWC? Which features of the model are most important in cases of firn aquifer presence?

**Minor and techical comments**

**Abstract**

As you redefine all acronyms again later in the text, I think it would be better if the abstract did not contain acronyms. This will help make the abstract more clear and approachable.

L5 – "to approximate a firn model" I would recommend being more specific here since your emulator does not approximate the entire firn model but merely a part of it.

L11 – "For SSP5-8.5, PFAs expand to Ellsworth Land in West Antarctica and Enderby Land in East Antarctica" – This is only true for some cases of SSP5-8.5 right?

L12 – "For climatic forcings from RACMO and MAR, we find that liquid water input (melt and rain) and snow accumulation are good predictors for PFA occurrence." – How do you quantify this? I would recommend computing some feature importance metrics to quantify which features are the most important in your emulator.

L14 – specify "air" temperature.

**Introduction**

L20 – "reduces their buttressing effect" → "reduces *ice-shelf* buttressing effect"

L20 – "Another process to reduce the…" → "Another process that can potentially reduce the…". I think the phrasing of this sentence should be modified a bit because the process of hydrofracture does not directly impact ice-shelf buttressing. Hydrofracture has more of an indirect effect because it can impact the ice-shelf volume (i.e by causing ice-shelf disintegration). This change in ice-shelf volume then reduces ice-shelf buttressing.

L25 – Maybe it would be better to write this as: "Currently, 60% of the ice-shelf area buttresses upstream ice and…"

L28 – "Perennial firn aquifers…, potentially causing ice shelves to break up". Please cite Montgomery et al 2020 or the Firn on Ice Sheets review paper (https://doi.org/10.1038/s43017-023-00507-9) here.

L42 – Fix "climaticx"

L43 – I think the statement "The absence of ice shelves along most of the north-western AP coast also suggests that the combination of ice shelves and firn aquifers is not viable" is misleading… There is no evidence that suggests that the NW AP has no ice shelves because of PFAs (which is what is indicated by this statement). Instead, the atmospheric and oceanic conditions on the NW side of the AP contribute to the lack of ice shelves. It may be that the climatic conditions which make ice shelves unviable are the same as those which allow PFAs to form. I find this statement misrepresentative as currently written.

L60 – add "properties" after "firn"

**Methods**

Some details in the IMAU-FDM firn modeling section are missing. What sort of spin-up do you do for the firn model? How thick are the firn layers in your model? Are they add steady state when you begin your simulations?

L95 – "see next section" → "(see section 2.2.1)"

L123 – "These are the most…" → "These input features describe the most…"

L125 – "For the summer, we consider snowfall instead of snow accumulation due to the likely presence of evaporation along side sublimation, which has different implications for PFAs". I don't follow this reasoning for including DJF snowfall, but annual and MAM snow accumulation. Why does this need to be different for different periods of the year?

L130 – How many total input features are there then for your model?

Section 2.2.3 – Which parameter is optimized? R2? RMSE?

**Emulator tuning and evaluation**

L 178 – This reference should be Table 2
L179 – Do you also leave out the 30-year rain input feature?

L184 – I think the R2 value here should be 0.89.

L187 – What is meant by "the individual locations" in this sentence?

L188 – "The performance is poorest on and around the Larsen C and D ice shelves…" This makes sense to me because you are asking the model to extrapolate to warmer climate conditions that are likely not seen during the training which utilizes lower emission scenarios.

**Results**

L226 – "Notably, HIRHAM-EC-Earth predicts at least twice the initial (2015)…" Is EC-Earth much warmer in the present climate? Higher precipitation?

L242 – "For example, on Larsen C ice shelf most aquifers are predicted by MAR…" What is meant by "most aquifers" here. "Most" compared to what?

Section 4.4 – Why are some firn aquifers transient? From the snow modeling output, what happens to make the aquifers disappear?

L291 – "As future warming leads to increased melt accumulation, the emulator is expected to produce more accurate PFA predictions..." This seems highly speculative and I don't fully understand the logic behind this statement.

**Discussion**

L325 – CESM2 also has a high precipitation bias

L331 – "Thus, the importance of including PFAs when assessing the timing of ice-shelf vulnerability also decreases" Can you elaborate on this? I don't fully understand.

**Figures**

Figure 2 c/d – The x and y-labels are "Years with PFAs" but the units are "mm".

---

## Referee Comment (RC2)

**Review: "Emulating the future distribution of perennial firn aquifers in Antarctica"**

by Veldhuijsen et al.

Submitted to *The Cryosphere*

**1   General**

In this paper, the authors analyze the formation of perennial firn aquifers in Antarctica under future emission scenarios using a machine-learning emulation of a firn model. They use several climate models and show the distribution of resulting firn aquifer, mostly as a total number of years with a firn aquifer until 2100. There are considerable differences in the predictions based on the climate models, which confounds the interpretation. The paper is straightforwardly constructed and presents the results factually. The paper lacks a central message and the title reflects the factual read out of results. I am not opposed to this paper being published and I submit a few ideas / questions in the sections below.

**2   Remarks**

1. SSP5-8.5 is (currently) an unlikely emission scenario, so I think it could be worth providing some context.

2. Although firn aquifer can form in certain conditions, they only appear in a few regions. There are many reasons that might prevent PFAs from developing in these places, e.g. ice lens, surface rives, or hydrofracture.

3. The distribution of firn aquifers in Antarctica has been computed already, and the climate models are uncertain, so I am curious what we learn from the emulation. I agree that it is a less computationally expensive way to perform the simulations.

4. I am not entirely convinced by the data in figure 2 suggesting that there is a significant spread between the firn model and the emulation.

**3   Specific comments**

1. Figure 5 could have more location labels, especially the Enderby Land part. A similar comment applies to all of the maps.

2. Figure 5: there are pretty significant differences in the location and duration of PFAs for the different climate models. How should we interpret this uncertainty?

3. Acknowledgements: in which sections was ChatGPT used?

---

## Author Response (AR1)

**Response to Community Comment:**

First of all, we would like to thank Irina Overeem and the graduate seminar group at the University of Colorado for their time for reviewing our work. We appreciate the constructive feedback we received. Following the feedback we have clarified the text. Below, we address your specific suggestions about the introduction, organization and discussion the manuscript. Responses to the comments of the reviewers are written in red and citations of the manuscript are written in blue.

**Kind regards, Sanne Veldhuijsen**

**Notes for improvement:**

1. In the introduction, we felt that the motivating or overarching purpose of the paper was unclear to the broader cryosphere audience. More time was spent explaining known instability on ice sheets in Antarctica (which is good) but we feel that it would be helpful to add more information and context of why PFAs are important.

In this context it may help to clarify the link between PFAs and known processes happening in ice sheets. More evidence towards the importance and function of PFAs, would help solidify the purpose of this paper.

Thank you for this comment. We agree that more information can be added about the importance of PFAs.

**2nd paragraph of the introduction:**

"While much attention has been directed to melt accumulation on the ice shelf surface following firn air depletion, meltwater storage within the firn layer can also modulate meltwater drainage to crevasses (e.g. Scambos et al. 2009). Under climatic conditions characterized by high surface melt rates and snow accumulation, liquid water can persist year-round within the firn's pore space, forming perennial firn aquifers (PFAs). As snow has a low thermal conductivity (Calonne et al. 2019) high snowfall rates in fall and winter rapidly cover and insulate the summer meltwater and thereby prevent it from refreezing in winter."

3rd paragraph: general info about contemporary PFA aquifer distribution (also Greenland).

**4th paragraph of the introduction:**

"Firn aquifers provide sufficient water storage capacity to contribute to ice shelf disintegration events, as evidenced by the partial break-ups of the northern and northwestern Wilkins ice shelf in 1993, 1998, and 2008–2009, where the detection of bright reflectors in airborne radar surveys indicated the presence of a firn aquifer (e.g. Braun et al., 2009, Scambos et al., 2000, Montgomery et al. 2018). The year-round availability of water at depth can lead to hydrofracturing, when stresses conditions shift to favor tensile extensions or reduced compression, even during the winter (Scambos et al., 2009). This mechanism can initiate cascading drainage events that cause rapid and large-scale disintegration, as observed for the Wilkins ice shelf in 2008 (Scambos et al. 2009)."

2. Similarly, the paper seems to be method focused, but the motivation for these methods was not clearly defined within the introduction.

We address this as follows: "Therefore, we developed an XGBoost PFA emulator. Emulators are statistical or machine learning models that are trained on specific output of a more complex model. Emulators can be based on simple statistical models, such as linear regression, or use more advanced machine learning approaches, such as Random Forest or XGBoost regressors.

Several studies have used emulators to simulate firn properties (Dunmire et al., 2024; Verjans et al., 2021; Jourdain et al., 2024), as well as other ice sheet processes (Van Katwyk et al 2023). XGboost is a decision tree-based method (Chen et al. 2016) that has been shown to outperform other machine learning algorithms, such as neural networks, random forest regression and linear regression methods, for various glaciological applications (Anilkumar et al., 2023; Iban et al., 2023; Sun et al., 2024)."

- 3. A minor note is that reducing the use of acronyms may be beneficial to people who are not as familiar with this topic and the subsections of Antarctica. Redefining the major acronyms at the beginning of new sections would help reinforce these terms for the reader.

  Thank you for this suggestion, we will incorporate that throughout the manuscript.
- 4. We believed that this was a predominantly methods-focused paper, but we felt that the motivation for these methods was not clearly defined within the introduction, especially for a broader audience in the cryosphere community. While we liked that time was spent explaining known instability on Antarctic ice sheets, we thought that it would be helpful to add more context for why PFAs are important. As a start, we would like to see a more clearly defined link between PFAs and known processes in ice sheets. Another minor suggestion is to reduce the use of acronyms may be beneficial to people who are not as familiar with this topic and the subsections of Antarctica. Redefining the major acronyms at the beginning of new sections would help reinforce these terms for the reader.

Thank you for these suggestions, see my replies to comment 1, 2 and 3.

5. The methods section could benefit from a clearer explanation of the data flow behind the development of the machine learning algorithm with use of the IMAU-PFA model, and its application (and the climate models used as input). This could take the form of transition sentences between individual paragraphs to emphasize which results are emulated or modeled, which are just input data. In addition, a workflow diagram can clarify this process, especially for those unfamiliar with the range of available climate models. We think it could be made more clear when data is presented in figures and otherwise are clearly labeled as input or modeled versus emulated results. Such clarification would also serve to make the contribution of the emulator clearer.

To improve the clarity, we will specify whether the results are from a climate model directly or simulated with the emulator, also in the figure caption, and especially in the transition sentences.

6. Again in the discussion a few notes on the broader context and significance may help. While this paper/study demonstrates the projected expansion of PFAs under different climate scenarios, it would benefit a cross section of the cryosphere community with a more comprehensive discussion linking the expansion of PFAs to the stability of the Antarctic ice sheet and broader climate impacts (i.e. sea-level rise).

We decided to keep our paper concise and focus on the future distribution of PFAs, thereby indicating which ice shelves potentially become vulnerable to hydrofracturing, but we do not elaborate on their specific impact on future sea level rise. The proposed broader discussion is in our view appropriate for a review paper, however our manuscript is not aiming to be a review paper.

7. We did appreciate Section 5.4 as being succinct. It briefly explores the transient life cycle of PFAs and their implications for ice sheet stability, but it could benefit from a clearer connection to the study's findings. More explicitly linking the formation, expansion, and eventual

depletion of PFAs to potential impacts on specific ice shelves - under varying accumulation and warming conditions (different climate models) - could strengthen the discussion.

We refer back to Figure S4 from the supplementary materials to show on which specific ice shelves transient aquifers are simulated: "Our results only show transient PFAs along the boundaries of PFA regions and on Wilkins ice shelf in SSP5-8.5, but this is expected to increase after 2100." Furthermore, the relation between the modelled climate change and potential transient PFA occurrence is already discussed in Section 4.4. We don't see the need to repeat this discussion here.

8. We suggest that situating PFAs within the context of ice-sheet mass balance and the climate system could help highlight why understanding PFA distribution is a critical component in forecasting future ice-sheet stability and sea-level implications.

See our explanation to comments 1 and 6.

**New references**

Scambos, T., Fricker, H. A., Liu, C. C., Bohlander, J., Fastook, J., Sargent, A., ... & Wu, A. M. (2009). Ice shelf disintegration by plate bending and hydro-fracture: Satellite observations and model results of the 2008 Wilkins ice shelf break-ups. *Earth and Planetary Science Letters*, 280(1-4), 51-60.

Braun, M., Humbert, A., & Moll, A. (2009). Changes of Wilkins Ice Shelf over the past 15 years and inferences on its stability. *The Cryosphere*, 3(1), 41-56.

**Response to Reviewer 1:**

First of all, we would like to thank Devon Dunmire for the time for reviewing our manuscript. We appreciate the constructive and insightful feedback we received. Following the feedback we have improved the explanation of our methodology and the clarity of the text. Below, we address your specific suggestions. Responses to the comments of the reviewers are written in red and citations of the manuscript are written in blue.

**Kind regards, Sanne Veldhuijsen**

**Major comments**

First, L80 reads: "The bucket method is computationally efficient but does not allow for saturated pore spaces, preferential flow, standing water over ice layers, or horizontal flow". Since a firn aquifer is quite literally "saturated pore space", I think a more in depth explanation about why the bucket scheme can be used in this work is required.

Thank you for this suggestion. We already elaborated on this in the discussion, but we add an explanation: "Since PFAs are defined as saturated firn, this is something different from what IMAU-FDM can simulate. However, the model does simulate the insulation of downward percolating liquid water by sufficiently high accumulation rates, that prevent the meltwater from refreezing in the winter. Therefore, the presence of year-round or spring liquid water in the firn has been successfully used to identify the presence of aquifers (Forster et al. 2014, Kuipers Munneke et al. 2014, Van Wessem et al. 2018, Brils et al. 2024)."

And we already did elaborate on this in the discussion: "In addition, the use of irreducible water content hinders the ability to estimate the volume of meltwater stored within a PFA. ... While it is not possible to estimate the volume of meltwater stored within a PFA with the bucket method, the presence of year-round LWC in the firn can be used as an indication of PFA presence (Munneke et al. 2014). Brils et al. (2024) simulated aquifers in Greenland using IMAU-FDM, agreeing with 62 % of the observed aquifers by airborne and ground-penetrating radar measurements. The mismatch can be explained by. ...."

The paragraph in lines 115-119 is very confusing. I don't understand the reasoning for why it is necessary to introduce a negative value to counterbalance the positive value. Also how are these values 'of the same order of magnitude'? In Figure 2a,b there are many points with LWC above 1000; however I would think the minimum possible value for the number of days without LWC is -365. When you say "we introduce a negative value...", what is meant by this? Do you average this value with the perennial LWC? Or is your model then predicting 2 separate target variables?

Thank you for these concerns. We agree that this explanation can be confusing and lacks some information, and therefore add more explanation about the rationale behind this approach. We rephrase this entire paragraph as follows: "Our target variable is the annual perennial LWC, which is defined as the minimum vertically integrated LWC over a year, based on model output at 10-day intervals. A yearly perennial LWC of zero indicates the absence of LWC at some point during a year, meaning there is no PFA. A drawback of this approach is that it does not differentiate between years with brief periods without LWC (e.g. 10 days) and years with long periods without LWC (e.g. 10 months). Since the target variable, the annual perennial LWC, cannot take negative values, this prevents negative values from counterbalancing positive biases in the predictions (e.g. an asymmetric distribution of the errors). To address this, we introduce negative values to represent the number of days without LWC during periods when PFAs are absent. This allows us to capture both the presence and absence of LWC in a unified way. For example, if LWC is absent for 10 days, the target value is -10; if LWC is absent for

10 months, the target value is approximately -300. Although the negative and positive quantities are conceptually different, their absolute magnitudes are comparable in scale, which provides a relatively smooth transition around zero. To further clarify, this method does not involve averaging LWC values with negative values. Instead, the target variable is a single, unified measure that can take either positive values (indicating LWC amount) or negative values (indicating days without LWC). If the target variable exceeds zero, it is considered to indicate the presence of a PFA."

Also, why can you not just predict the average LWC during the winter months, instead of the annual minimum? As winter is the least-likely month for liquid water to exist, maybe this would be a simpler way to predict PFA presence?

In some studies, the presence of LWC in winter or spring is indeed used to identify PFAs. However, in some regions winter melt events occur, so to minimize the risk of classifying this as a PFA, we use the minimum amount of LWC during a year. Also, if for one month/week there is no liquid water, it means the aquifer is not perennial. Therefore, this approach is a more correct way to identify PFAs.

Finally, from the target variable, what do you actually consider to be a PFA? This is unclear from the text. We clarify this with the following sentence: "If the target variable exceeds zero, it is considered to indicate the presence of a PFA."

Further, in section 4 (L184): It does not make sense to me to report the RMSE as 86 mm or days. I think I just don't understand the target variable but I believe it to be some combination of LWC and days without PFA presence it which case the RMSE would not strictly be 86mm or 86 days but some combination of the two?

We hope that our explanation to the comments above regarding the target variable helps to clarify this point.

Finally, a look into feature importance would be a very valuable addition to this work. Are there features that are more heavily utilized by the model than others for predicting LWC? Which features of the model are most important in cases of firn aquifer presence?

The list of permutation importance metrics is shown below. However, a part of our training dataset, especially for SSP1-2.6 and SSP2-4.5 represents conditions that are not conducive to aquifer formation (e.g. relatively cold regions). As a result, some features might have a high importance, just because they exclude those regions. For features like aquifer presence, we therefore think the permutation importance is not meaningful as it primarily shows which parameters are needed to predict the presence of a PFA, not its magnitude. This is in contrast to e.g. firn air content, which is non-zero for most conditions, and subsequently the feature importance highlight the parameters relevant for the FAC.

**Minor and technical comments**

**Abstract**

As you redefine all acronyms again later in the text, I think it would be better if the abstract did not contain acronyms. This will help make the abstract more clear and approachable.

We remove the acronyms from the abstract except for PFAs, and also redefine some of the major acronyms at the beginning of new sections.

L5 – "to approximate a firn model" I would recommend being more specific here since your emulator does not approximate the entire firn model but merely a part of it.

Indeed, we suggest rephrasing the following text: "To overcome this, we developed an XGBoost emulator, a fast machine learning model, to approximate a firn model. The PFA emulator was trained with simulations from the firn densification model IMAU-FDM, forced by three emission scenarios (SSP1-2.6, SSP2-4.5 and SSP5-8.5) of the combined regional climate model (RCM) RACMO2.3p2 and general circulation model (GCM) CESM2."

into: "To address this, we develop an XGBoost perennial firn aquifer emulator, a fast machine learning model that is trained on PFA output of simulations from the firn densification model IMAU-FDM. The firn simulations were forced by the combined regional climate model RACMO2.3p2 and general circulation model CESM2 for three emission scenarios (SSP1-2.6, SSP2-4.5 and SSP5-8.5)."

L11 – "For SSP5-8.5, PFAs expand to Ellsworth Land in West Antarctica and Enderby Land in East Antarctica" – This is only true for some cases of SSP5-8.5 right?

For completeness, we rephrase this as follows: "For SSP5-8.5, PFAs expand to Ellsworth Land in six out of the seven simulations and to Enderby Land in East Antarctica in five out of the seven simulations."

L12 – "For climatic forcings from RACMO and MAR, we find that liquid water input (melt and rain) and snow accumulation are good predictors for PFA occurrence." – How do you quantify this? I would recommend computing some feature importance metrics to quantify which features are the most important in your emulator.

To clarify, we rephrase the following text: "For climatic forcings from RACMO and MAR, we find that liquid water input (melt and rain) and snow accumulation are good predictors for PFA occurrence. However, HIRHAM predicts considerably less surface melt and accumulation for a given air temperature than MAR and RACMO do, resulting in less realistic PFA predictions."

into: "The emulator results for RACMO and MAR agree on the snowmelt and accumulation conditions required for PFA formation. While these results for HIRHAM are slightly different, caused by different modelled relations between temperature, accumulation and melt compared to RACMO."

This is also rephrased in the conclusion section.

L14 – specify "air" temperature. Done. Thank you for noticing this.

**Introduction**

L20 – "reduces their buttressing effect" → "reduces ice-shelf buttressing effect" We have changed this accordingly.

L20 – "Another process to reduce the..."  $\rightarrow$  "Another process that can potentially reduce the...".

I think the phrasing of this sentence should be modified a bit because the process of hydrofracture does not directly impact ice-shelf buttressing. Hydrofracture has more of an indirect effect because it can impact the ice-shelf volume (i.e by causing ice-shelf disintegration). This change in ice-shelf volume then reduces ice-shelf buttressing.

We agree and clarify this as follows:

"Another process that can potentially indirectly reduce the buttressing effect of ice shelves is melt pond-driven hydrofracturing. This mechanism contributes to ice-shelf disintegration thereby reducing the ice-shelf volume, which in turn reduces the ice-shelf buttressing. This process is expected to increase under future warming (Lai et al. 2020, van Wessem et al. 2023)"

L25 – Maybe it would be better to write this as: "Currently, 60% of the ice-shelf area buttresses upstream ice and..."

We agree and have adjusted this accordingly.

L28 – "Perennial firn aquifers..., potentially causing ice shelves to break up". Please cite Montgomery et al 2020 or the Firn on Ice Sheets review paper (https://doi.org/10.1038/s43017-023-00507-9) here.

Done, see our response to the Remark 1 of the Community comment.

L42 – Fix "climaticx" Done.

L43 – I think the statement "The absence of ice shelves along most of the north-western AP coast also suggests that the combination of ice shelves and firn aquifers is not viable" is misleading... There is no evidence that suggests that the NW AP has no ice shelves because of PFAs (which is what is indicated by this statement). Instead, the atmospheric and oceanic conditions on the NW side of the AP contribute to the lack of ice shelves. It may be that the climatic conditions which make ice shelves unviable are the same as those which allow PFAs to form. I find this statement misrepresentative as currently written.

In hindsight, the lack of ice shelves could indeed also be caused by the oceanic conditions. Therefore, we decided to remove this statement.

L60 – add "properties" after "firn" Done.

**Methods**

Some details in the IMAU-FDM firn modeling section are missing. What sort of spin-up do you do for the firn model? How thick are the firn layers in your model? Are they add steady state when you begin your simulations?

We add more details about the IMAU-FDM simulations: "An initial firn layer is obtained by looping over the forcing of the 1950-2000 reference period until the firn layer is in equilibrium with the surface climate. The model employs up to 3000 layers of 3 to 15 cm thickness, which represent the firn properties in a Lagrangian way."

L95 – "see next section"  $\rightarrow$  "(see section 2.2.1)" Done

L123 – "These are the most..." → "These input features describe the most..."

**Done**

L125 – "For the summer, we consider snowfall instead of snow accumulation due to the likely presence of evaporation along side sublimation, which has different implications for PFAs". I don't follow this reasoning for including DJF snowfall, but annual and MAM snow accumulation. Why does this need to be different for different periods of the year?

This statement was indeed not clearly formulated. Liquid and solid evaporation is largest during summer due to the higher atmospheric temperatures. However, we would like to include only how much the surface firn is replenished by new snow during the summer, as this could influence the overall densification of the firn layer and the evolution of PFAs. For the fall months and the annual averages, the net accumulation is deemed more relevant. We didn't test this, however, in detail. We reformulate this sentence to: "For summer, we consider snowfall instead of snow accumulation due to the larger magnitudes of liquid and solid evaporation. This is considered to better capture the summer replenishment of firn air content and hence pore space for PFAs."

L130 – How many total input features are there then for your model?

In total, the emulator requires 23 input features. 5, 10 and 30 year data for 7 atmospheric variables in combination with elevation and slope. We add: "This amounts to 23 input features in total."

Section 2.2.3 – Which parameter is optimized? R2? RMSE?

We optimize based on R2, this is clarified as follows: "Firstly, we evaluate the performance using spatial blocking cross-validation, with R2 as the scoring metric."

**Emulator tuning and evaluation**

L178 – This reference should be Table 2 Thank you for noticing this, done.

L179 – Do you also leave out the 30-year rain input feature?

Yes, to clarity this, we have replaced this: "30-year melt input feature" by: "30-year total annual liquid water input (surface melt plus rainfall) input feature"

L184 – I think the R2 value here should be 0.89.

Indeed, this has been updated.

L187 – What is meant by "the individual locations" in this sentence?

To clarify, we change this: "The emulator yields an R2 value of 0.90 for the years with PFAs of the individual locations (Fig. 2c)."

Into: "The emulator yields an R2 value of 0.90 for the years with PFAs of each grid point (Fig. 2c)."

L188 – "The performance is poorest on and around the Larsen C and D ice shelves..." This makes sense to me because you are asking the model to extrapolate to warmer climate conditions that are likely not seen during the training which utilizes lower emission scenarios. These results are for the spatial blocking evaluation. Region 7 has a distinct climate in terms of its relatively high melt and rather "low" accumulation. To avoid this confusion, we add this, so the sentence will be changed into: "The performance, while using spatial blocking, is poorest on and around the Larsen C and D ice shelves..."

**Results**

L226 – "Notably, HIRHAM-EC-Earth predicts at least twice the initial (2015)..." Is ECEarth much warmer in the present climate? Higher precipitation?

Yes, this was already explained in L236 (Section 4.3 Climatic drivers): "For HIRHAM-EC-Earth, the initial temperature is high, related to the warm bias over Antarctica in EC-Earth3 (Boberg et al., 2022), which explains the high PFA extent at the start of the simulation." We add a reference to Section 4.2 at this point: "Notably, HIRHAM-EC-Earth predicts at least twice the initial (2015) PFA extent (72,000 km2) compared to the other simulations, related to the higher initial temperatures (see Sec. 4.3). "

L242 – "For example, on Larsen C ice shelf most aquifers are predicted by MAR..." What is meant by "most aquifers" here. "Most" compared to what?

To clarify we rephrase this: "For example, on Larsen C ice shelf most aquifers are predicted by MAR, which is related to high surface melt and high accumulation at the foot of the mountains, which are absent in the other two models."

As: "For example, on Larsen C ice shelf the emulator predicts more aquifers for MAR compared to RACMO and HIRHAM, which is related to high surface melt and high accumulation at the foot of the mountains, which are absent in the other two regional climate models."

Section 4.4 – Why are some firn aquifers transient? From the snow modeling output, what happens to make the aquifers disappear? We add a more physical explanation for this: "As Figure 4b shows, PFAs can also develop, shrink and subsequently disappear, henceforth referred to as transient PFAs. This shrinkage and disappearance occurs when the firn layer becomes too thin (i.e. the firn air too depleted) to host an aquifer."

L291 – "As future warming leads to increased melt accumulation, the emulator is expected to produce more accurate PFA predictions..." This seems highly speculative and I don't fully understand the logic behind this statement.

To clarify, we rephrase this: "As future warming leads to increased melt and accumulation, the emulator is expected to produce more accurate PFA predictions, as more locations will shift into distinct PFA climate regimes rather than remaining in transitional states."

As: "As future warming leads to increased melt and accumulation, the emulator is expected to produce more accurate PFA predictions compared to the contemporary estimates. This is because warming is likely to shift more locations into well-defined PFA climate regimes characterized by high melt and accumulation, where aquifer formation is relatively certain, rather than into regimes with lower melt and accumulation, where aquifer formation is less certain (Fig. 11)."

**Discussion**

L325 – CESM2 also has a high precipitation bias

This is now included, and another comment is added: "However, these biases do not necessarily persist in the downscaled RCM output (Veldhuijsen et al. 2024)."

L331 – "Thus, the importance of including PFAs when assessing the timing of ice-shelf vulnerability also decreases" Can you elaborate on this? I don't fully understand.

To clarify, we propose to change this: "The duration of such transient PFA presence will decrease with stronger warming rates or lower accumulation rates. Thus, the importance of including PFAs when assessing the timing of ice-shelf vulnerability also decreases."

Into: "The duration of such transient PFA presence will decrease with stronger warming rates or lower accumulation rates due to relatively quick firn air depletion. Thus, the difference in the timing of ice-shelf vulnerability because of PFAs and firn air depletion becomes smaller."

**Figures**

Figure 2 c/d – The x and y-labels are "Years with PFAs" but the units are "mm". Changed.

**Response to Reviewer 2:**

First of all, we would like to thank the reviewer for the time for reviewing our work. We appreciate the constructive feedback we received. Following the feedback we have improved the explanation of our methodology, the clarity of the text and we highlighted the implications of our work. Below, we address your specific suggestions. Responses to the comments of the reviewers are written in red and citations of the manuscript are written in blue.

**Kind regards, Sanne Veldhuijsen**

**General**

In this paper, the authors analyze the formation of perennial firn aquifers in Antarctica under future emission scenarios using a machine-learning emulation of a firn model. They use several climate models and show the distribution of resulting firn aquifer, mostly as a total number of years with a firn aquifer until 2100. There are considerable differences in the predictions based on the climate models, which confounds the interpretation. The paper is straightforwardly constructed and presents the results factually. The paper lacks a central message and the title reflects the factual read out of results. I am not opposed to this paper being published and I submit a few ideas / questions in the sections below.

**2 Remarks**

1. SSP5-8.5 is (currently) an unlikely emission scenario, so I think it could be worth providing some context.

We agree that SSP5-8.5 is considered an unlikely emission scenario, and we have provided context regarding this in the methods Section 2.3 Predicting firn aquifers: "Although the probability of SSP5-8.5 scenarios is conceived as low, they are useful to assess low-probability/high-impact futures, and it characterizes the climate of the 22nd century (the probability for these conditions increases to about 7% by 2150) (Sarofim et al. 2024). In addition, we are limited by the unavailability of downscaled forcing datasets over Antarctica for the alternative emission scenario SSP3-7.0."

2. Although firn aquifer can form in certain conditions, they only appear in a few regions. There are many reasons that might prevent PFAs from developing in these places, e.g. ice lens, surface rives, or hydrofracture.

Yes indeed, we agree. We add more details about this in the discussion Section 5.1 Firn model and observations, (underlined indicate the additions):

"High surface melt and snowfall conditions are favorable for aquifer formation, however some processes, which are not included in IMAU-FDM can inhibit their formation despite these conditions. For instance, firn aquifers are known to drain in heavily crevassed areas (Cicero et al., 2023; Poinar et al., 2017), a process that is not included in the firn model. In addition, ice lenses can also limit vertical meltwater percolation to deeper firn, thereby increasing refreezing (e.g. Culberg et al. 2021), although they generally form under lower accumulation conditions (Brils et al. 2024). The bucket method does not include preferential flow, which can lead to an underestimation of the percolation depth of liquid water and therefore aquifer recharge as observed in Greenland (Miller et al., 2018; Vandecrux et al., 2020). The firn model also does not allow for lateral meltwater transport, which has been observed within aquifers, but with a limited measured specific discharge (60 - 140 m/yr) (Montgomery et al. 2020, Miller et al. 2018). On the other hand, not including lateral meltwater transport in streams can also result in underestimating aquifer presence. For instance, on the firn-terminating Priestley Glacier, meltwater flows down the ice shelf across blue ice into the section of the ice shelf with thicker surface snow, where it probably saturates the firn and feeds firn aquifers (Bell et al. 2017).

Similar firn-terminating streams have been observed on the Amery and Nivilsen ice shelves, as well as in Dronning Maud in East Antarctica (e.g. Fricker et al. 2002, Kingslake et al. 2015, Lepparanta et al. 2013)."

3. The distribution of firn aquifers in Antarctica has been computed already, and the climate models are uncertain, so I am curious what we learn from the emulation. I agree that it is a less computationally expensive way to perform the simulations.

The distribution of firn aquifers has only been computed over the contemporary climate for the Antarctic Peninsula (Van Wessem et al. 2018, Di Biase et al. 2024), so the future distribution was still unexplored. The exploration presented here is impossible without the emulators, as running IMAU-FDM for all simulations is very costly, and, moreover, the required input data is not available. We emphasize this better in the introduction, by adding the underlined sentence to this part:

"Increasing surface melt and snowfall may result in future PFA expansion, also to other regions of Antarctica. Similarly, an inland expansion of aquifers occurred over the last decades in Greenland (Horlings et al., 2022). However, the future distribution of PFAs in Antarctica has not yet been explored."

**and**

"However, running a firn model for multiple forcings is computationally demanding and requires 3-hourly model output, which is regularly unavailable".

Moreover, we agree that we can articulate what we learn from this work more explicitly, which is mainly, that Ellsworth land and Enderby Land region are also prone to firn aquifer formation under strong warming scenarios (or intermediate warming after 2100). We rephrase the main findings as two additional paragraphs in Section 5.4 Implication for future ice-shelf stability: "While there is a substantial spread in PFA predictions by the emulator between the climate models, some general patterns emerge. Under SSP1-2.6 and SSP2-4.5, the emulator predicts that PFAs remain restricted to, but expand within, the AP, the region where aquifers are currently present. For SSP5-8.5, aquifer expansion is more pronounced within the AP. In addition, for SSP5-8.5, PFAs expand to Ellsworth Land in six out of the seven simulations and to Enderby Land in East Antarctica in five out of the seven simulations.

This expansion within the AP can potentially contribute to ice-shelf collapse in this region, also for SSP1-2.6 and SSP2-4.5. Significant tensile stresses observed on the Wilkins and Larsen C ice shelves (Lai et al., 2020) suggest that these areas may be vulnerable to further instability caused by PFAs. Although the PFA expansion on Larsen C ice shelf by MAR might be overestimated, as explained in Section 5.3. The stress regime on George VI suggests it is currently resilient to hydrofracturing. In Ellsworth Land, substantial tensile stresses on the Abbot ice shelf (Lai et al., 2020) suggest that PFAs under SSP5-8.5 could potentially induce hydrofracturing here. Similarly, ice shelves in Enderby Land are vulnerable to collapse, though they do not provide substantial buttressing (Lai et al., 2020). Furthermore, PFA expansion on the grounded ice, which mainly occurs in the northwestern AP, could cause meltwater drainage through crevasses to the bed (Poinar et al., 2017). This drainage may result in enhanced basal lubrication, increasing ice velocity and ice discharge into the ocean (Zwally et al., 2002)."

We also add this sentence after the last paragraph of this section: "However, the onset of the aquifer is still decades earlier than the firn air depletion, highlighting the importance of considering PFAs, rather than solely firn air depletion, when assessing vulnerability to hydrofracturing."

4. I am not entirely convinced by the data in figure 2 suggesting that there is a significant spread between the firn model and the emulation.

The figure indeed suggests that there is a spread between the firn model and the emulation. The reason for this, is that the emulator is sometimes a few years earlier or later in predicting the onset of the aquifer. However, if we look at Figure 3. This shows that although there are differences, the firn model and its emulation agree in regions where aquifers are simulated, and that the differences are rather subtle. This is emphasized in Section 3.2 (underlined is added): "The emulator yields an R² value of 0.90 for the years with PFAs of the individual locations (Fig. 2c). Maps of PFA years of the AP and Ellsworth Land for the spatial blocking for SSP5-8.5 are shown in Fig. 3. Although Figure 2c indicates that there is some spread and outliers, Figure 3 demonstrates that the firn model and emulator largely agree on the firn aquifer distribution, and the differences are rather subtle."

**3 Specific comments**

1. Figure 5 could have more location labels, especially the Enderby Land part. A similar comment applies to all of the maps.

We have dashed boxes in Figure 1 to indicate the location of the specific regions. I will refer to Figure 1 in the caption of each Figure to clarify this.

2. Figure 5: there are pretty significant differences in the location and duration of PFAs for the different climate models. How should we interpret this uncertainty?

Indeed, there is a significant spread in PFA prediction between the global and regional climate models. For SSP5-8.5 this is mainly caused by the large spread in warming shown by the different warming trends in Fig. 9b. Similarly, for Greenland by Glaude et al. (2024) found a factor two difference in 21st century Greenland ice sheet surface mass balance projections from three regional climate models driven by the same climate model. But in this work, for SSP1-2.6 and SSP2-4.5 the differences in PFA projections are also substantial, which is more reflected by the difference in the contemporary climate (Fig.10). All in all, this shows that the exact PFA predictions are uncertain because the magnitude and regional patterns of climate change are uncertain. But still there are general patterns that we see, from which we can learn, in terms of regions that are most prone to PFA expansion. And this also highlights the usefulness of an emulator, as we can incorporate a wide range of climate models. We incorporate this in the discussion as, first at the start of Section 5.3: "The emulator predicts a large range of possible PFA distributions, caused by differences in the climatic forcing. This highlights the usefulness of the emulator as it allows incorporating a wide range of climate forcings to better capture the variability and uncertainty."

And in Section 5.4 (in addition to the revision based on comment Remark 3): "While there is a substantial spread in PFA predictions by the emulator between the climate models, some general patterns emerge."

2. Acknowledgements: in which sections was ChatGPT used? I don't remember which specific sections in which ChatGPT was used for text editing, so instead, I acknowledge its use in general terms: "SbmV used ChatGPT to enhance the readability of manuscript."

**New references**

Culberg, R., Schroeder, D. M., & Chu, W. (2021). Extreme melt season ice layers reduce firn permeability across Greenland. *Nature communications*, *12*(1), 2336.

Fricker, H. A., Allison, I., Craven, M., Hyland, G., Ruddell, A., Young, N., ... & Popov, S. (2002). Redefinition of the Amery ice shelf, East Antarctica, grounding zone. *Journal of Geophysical Research: Solid Earth*, 107(B5), ECV-1.

Kingslake, J., Ng, F., & Sole, A. (2015). Modelling channelized surface drainage of supraglacial lakes. *Journal of Glaciology*, 61(225), 185-199.

Leppäranta, M., Järvinen, O., & Mattila, O. P. (2013). Structure and life cycle of supraglacial lakes in Dronning Maud Land. *Antarctic Science*, 25(3), 457-467.

Sarofim, M. C., Smith, C. J., Malek, P., McDuffie, E. E., Hartin, C. A., Lay, C. R., & McGrath, S. (2024). High radiative forcing climate scenario relevance analyzed with a ten-million-member ensemble. *Nature communications*, 15(1), 8185.

Glaude, Q., Noël, B., Olesen, M., Van den Broeke, M., van de Berg, W. J., Mottram, R., ... & Fettweis, X. (2024). A factor two difference in 21st-century Greenland ice sheet surface mass balance projections from three regional climate models under a strong warming scenario (SSP5-8.5). *Geophysical Research Letters*, 51(22), e2024GL111902.

---

## Author Response (AR2)

**General Comment:**

1) The central message of the manuscript is still hard to determine in this revised version despite the new text providing more motivation and stronger conclusions from the presented results. Consider making the central message of the paper more explicit in the abstract and other relevant sections, and revising the title to reflect that central message/big takeaway from your results.

We agree and change the title to: "Emulating the expansion of Antarctic perennial firn aquifers in the 21st century."

**The abstract is rewritten as:**

"Perennial firn aguifers (PFAs) are year-round bodies of liquid water within firn, which modulate meltwater runoff to crevasses, potentially impacting ice-shelf and ice-sheet stability. Recently identified in the Antarctic Peninsula, PFAs form in regions with both high surface melt and snow accumulation rates, and are expected to expand due to the anticipated increase in surface melt and snowfall. Using a firn model to predict future Antarctic PFAs for multiple climatic forcings is relatively computationally expensive. To address this, we develop an XGBoost perennial firn aquifer emulator, a fast machine learning model. It was trained, using a scenario and spatial blocking evaluation approach, on PFA output of simulations from the firn densification model IMAU-FDM, which was forced by the combined regional climate model RACMO2.3p2 and global climate model CESM2 for three emission scenarios (SSP1-2.6, SSP2-4.5 and SSP5-8.5). The trained emulator was applied on nine additional forcings (2015-2100) from the regional climate models MAR and HIRHAM in combination with five global climate models. We show that the emulator is robust, explaining at least 89% of the variance in PFA presence and meltwater storage. Our results indicate that for the SSP1-2.6 and SSP2-4.5 scenarios, PFAs remain mostly restricted to the Antarctic Peninsula. For SSP5-8.5, PFAs expand to Ellsworth Land in six out of the seven simulations and to Enderby Land in East Antarctica in five out of the seven simulations. Furthermore, the emulator predicts PFAs for similar surface melt and accumulation conditions when forced with MAR or RACMO data. For HIRHAM these conditions are slightly different, due to the different relationship between temperature, accumulation and melt in HIRHAM compared to RACMO. Overall, our findings show that PFAs are likely to expand in a warmer Antarctica, irrespective of the emission scenario, increasing the risk that an ice shelf collapses due to hydrofracturing."

**The conclusion is rewritten as:**

"An XGBoost machine learning emulator was set up to predict future Antarctic perennial firn aquifers (PFAs) (2015-2100) for an ensemble of 12 simulations from three regional climate models (RCMs) (RACMO, MAR and HIRHAM) in combination with five global climate models (GCMs). The emulator was trained with simulations of three scenarios (SSP1-2.6, SSP2-4.5 and SSP5-8.5) from IMAU-FDM forced by RACMO-CESM. For training and testing, we used a scenario and spatial blocking evaluation approach and not random selections of training, testing and evaluation data, as the latter approach is vulnerable for overfitting. We firstly demonstrate that the emulator successfully estimates PFAs, it explains at least 89% of the variance in PFA presence and perennial liquid water content. Secondly, we project, based on the available model simulations, that for SSP1-2.6 and SSP2-4.5, PFA presence remains restricted to the Antarctic Peninsula (AP). For SSP5-8.5, PFAs expand within the AP, and expand to Ellsworth Land in six out of the seven simulations, and to Enderby Land in East Antarctica in five out of the seven simulations and to Marie Byrd Land in two out of the seven simulations. Lastly, we observe a large spread among the RCMs and

GCMs emulator predictions, related to differences in the climatic input, which highlights the necessity of analyzing multiple RCM and ESM combinations and usefulness of the emulator that make such analysis feasible. Further analysis of the results shows that the emulator projects, while using RACMO and MAR simulations, PFAs for similar surface melt and accumulation conditions. When the emulator is fed with HIRHAM data, these conditions are slightly different, caused by different relation between temperature, accumulation and melt in HIRHAM compared to RACMO. Conclusing, our results show that, irrespective of the emission scenario, firn aquifers are likely to expand in a warmer Antarctica. This highlights the importance to understand the impacts PFAs have on ice sheet hydrology, instability and ice-shelf instability."

Technical/Line-specific Comments:

\*line comments made with respect to line numbers on the track changes version of the manuscript

L11: no space needed between 89% (and every other percentage value reported in the manuscript).

Done, thank you.

L11-13: consider rearranging this sentence to avoid repeating 'Using' in two consecutive sentences: 'We predict future PFAs (2015-2100) using the PFA emulator for...'

We rephrase this as: "Using a scenario and spatial blocking evaluation approach, we found that the emulator successfully explains at least 89% of PFA presence and meltwater storage variance. We then apply the emulator to predict future PFAs (2015–2100) for nine additional forcings from the regional climate models MAR and HIRHAM in combination with five global climate models."

L13: consider using a description of the two emission scenarios you name here instead to allow the reader to quickly understand the context of those named scenarios. Something like: 'Under our low and moderate emissions scenarios (SSP1-2.6 and SSP2-4.5)...'

Done.

L13: consider the same approach for the next sentence too (SSP5-8.5) Done.

L14: you can remove 'the' from the added 'six out of the seven...'

Done, and also in other sections.

L15: like the previous sentence, you can remove 'the' from the added phrase here Done.

L15-21: consider merging these two added sentences to improve the flow. Something like: '...conditions required for PFA formation, while those for HIRHAM are lower due to the model's different relation between temperature, accumulation, and melt compared to RACMO.' Done.

L47: I think there's a missing word here: 'Combining regional climate model (RCM) output/data with satellite data...' We rephrase this as: "Combining regional climate model (RCM) output with satellite data confirms a high probability of PFA occurrence in all these regions."

L52-55: I appreciate the added text here to expand upon the description of the motivation here, but it might be further improved by outlining how 'water storage capacity' directly contributes to 'ice-shelf disintegration events' here faster. I think the easiest way to accomplish that is to put the information about the break-ups of the Wilkins ice shelf after the description of the mechanisms that the water impacts in Lines 56-59. We agree and have rephrased this paragraph as follows: "The year-round availability of water at depth in firn aquifers can lead to hydrofracturing when stress conditions shift to favor tensile extension or reduced compression, even during the winter (Scambos et al. 2009). This mechanism can initiate cascading drainage events that cause rapid and large-scale ice shelf disintegration. The partial break-ups of the northern and northwestern Wilkins ice shelf in 1993, 1998, and 2008–2009 could well exemplify this process, as the detection of bright reflectors in airborne radar surveys indicated the presence of a firn aguifer (Braun et al. 2009, Scambos et al. 2000, Montgomery et al. 2020). These events highlight how water stored in firn aquifers can potentially accelerate disintegration. Similarly, the Müller ice shelf, which also contains firn aguifers, has lost a considerable portion of its surface area (49%) since the 1950s (Cook et al. 2010) while neighbouring Jones and Wordie ice shelves, with similar climatic conditions, have completely disintegrated (Cook et al. 2010)."

L56: I think 'extensions' should be singular here Done.

L59-63: combine these references to ice shelf disintegration with the Wilkins ice shelf examples from Lines 53-55 here to improve flow. Done, see two comments above.

L64: a comma is needed after 'Greenland'. Done.

L78-79: I think you're missing a phrase at the end of this sentence here. Something like: 'Therefore, we developed an XGBoost PFA emulator to generate projections of future PFA extent with limited data.' We rephrase this as: "To enable projections under a wide range of climate forcings, we developed an XGBoost-based PFA emulator to estimate future PFA extent."

L99: either add 'just' before 'firn temperature' again here, change the wording to highlight that the updated densification rate depends on firn temperature like before but also grain size and overburden pressure. We rephrase this as: "The updated expression lets densification rate depend on firn temperature, grain size and overburden pressure, whereas previously densification rate depended on firn temperature along with averages over the past 40 years of accumulation and surface temperature."

L106-107: The wording in this added sentence is awkward. Consider rewording to something like "Since perennial firn aquifers (PFAs) are defined as saturated firn, IMAU-FDM cannot simulate them." Done.

L132: add 'because it' to the sentence here: '...and because it has been shown...' Done.

L136: there is a missing connecting phrase or word here between '...and complex models,' and 'it incorporates...' We rephrase this as: "Additionally, XGBoost is highly scalable, meaning it can handle large datasets and complex models. Furthermore, XGBoost

incorporates regularization techniques to prevent overfitting and it has built-in mechanisms to estimate feature importance."

L163: I think you mean 'ultimately' or 'also' here instead of 'therewith'. We rephrase this as: "These input features describe the most important mass fluxes and boundary conditions governing firn density, temperature and LWC, and ultimately the presence of PFAs."

L190: you can remove 'part of the' here. Done.

L230: while I think that your description of the target variable (annual perennial LWC) is much clearer, I still think that the reviewers' comments about reporting the RMSE and bias in mm and days being confusing, is still valid here. Can you include a phrase or sentence to quickly clarify that the mm units refer to the positive values and days refer to the negative values? We rephrase this as follows: "The RMSE is 86 (mm for positive values, days for negative), and the bias is 0.4 (mm for positive values, days for negative values)."

L238: two commas needed here: 'When randomly, instead of strategically, ...' Done.

L240: remove 'using' here. Done.

L251: 'onset of aquifers' \( \rangle \) 'onset of aquifer formation' Done.

4.1 Section title: 'PFA predictions...' Done.

L266: 'arises by' ◊ 'arises from' Done.

4.2 Section title: 'PFA predictions...' Done.

L276: add 'the' after 'yields'. Done.

L278: add 'to be' after 'predicted' Done.

L288: the original formulation of the sentence was correct: 'For example, MAR-NorESM predicts the smallest...' Adjusted accordingly.

L308: 'appear thus' \( \rightarrow \) 'thus appear' Changed.

L309: remove 'a' Done.

L313: '..., and henceforth will be referred to as...' Done.

L349-350: the phrase 'in addition to' here makes the sentence awkward and hard to interpret. Reword to clarify the sentence. We rephrase this as: "The mismatch can be explained by underestimation of observationally detected aquifers by e.g. widely spaced radar data, resolution and process limitations of the firn model, e.g. drainage of meltwater into crevasses that is not included in the model, other above-described firn model limitations, and/or biases in the climatic forcing."

L386: add 'in' after 'resulting' Done.

---

## Author Response (AR3)

Dear Editor,

We have incorporated the last two suggestions.

Thank you.

Sanne Veldhuijsen